# FedLog: Personalized Federated Classification with Less Communication and More Flexibility

**Haolin Yu**                                                                    *h89yu@uwaterloo.ca*
*University of Waterloo, Canada*
*Vector Institute for AI, Canada*
**Guojun Zhang***                                                       *guojun.zhang@uwaterloo.ca*
*Minimax, China*
**Hongliang Li**                                                          *hongliang.li2@huawei.com*
*Noah's Ark Lab, Huawei Technologies, Canada*
**Pascal Poupart**                                                           *ppoupart@uwaterloo.ca*
*University of Waterloo, Canada*
*Vector Institute for AI, Canada*

**Reviewed on OpenReview:** *https://openreview.net/forum?id=7HwkObvvKn*

## Abstract

Federated representation learning (FRL) aims to learn personalized federated models with effective feature extraction from local data. FRL algorithms that share the majority of the model parameters face significant challenges with huge communication overhead. This overhead stems from the millions of neural network parameters and slow aggregation progress of the averaging heuristic. To reduce the overhead, we propose FedLog, which shares sufficient data summaries instead of raw model parameters. The data summaries encode minimal sufficient statistics of an exponential family, and Bayesian inference is utilized for global aggregation. FedLog helps reduce message sizes and communication frequency. We prove that the shared messages are minimal sufficient statistics and theoretically analyze the convergence rate of FedLog. To further ensure formal privacy guarantees, we extend FedLog with the differential privacy framework. Empirical results demonstrate high learning accuracy with low communication overhead of our method.

## 1 Introduction

Representation learning plays a crucial role in machine learning by effectively extracting features from raw data to facilitate downstream predictions in various fields (Liu et al., 2024). Combined with federated learning (FL), federated representation learning (FRL) aims to learn personalized federated models and diminish the impact of heterogeneous clients. FRL methods separate a whole neural network into two parts, body and head (Liang et al., 2020; Collins et al., 2021; Arivazhagan et al., 2019). The body is a deep network that learns a compact feature representation from the raw data. The head is a shallow network with few layers that makes predictions in the representation space. Personalization is achieved by localizing either the body or the head. Parameters of the rest of the model are shared with the server for global aggregation with the FedAvg heuristic (McMahan et al., 2017).

FRL utilizing body parameter sharing faces two significant challenges. **1) Heavy communication overhead.** Sharing body parameters induces huge communication cost per round due to millions of parameters in the deep learning model (Wen et al., 2023) since the body contains the majority of the parameters. Furthermore, the averaging heuristic slows down the convergence with heterogeneous clients, resulting in more aggregation rounds. **2) Rigid model architecture.** Sharing body parameters requires the same body architecture among clients (McMahan et al., 2017; Arivazhagan et al., 2019; Collins et al., 2021; Zhang et al., 2024; Oh

---

*The work was done at their previous affiliation, Huawei Noah's Ark Lab.

et al., 2021; Li et al., 2021). However, clients usually possess different amounts of computing resources. Thus, one body architecture may not be suitable for all devices. In this case, clients with limited resources cannot effectively participate. Even though some recent works consider sharing head parameters to reduce communication overhead, their communication efficiency is still limited due to slow averaging (Karimireddy et al., 2020).

To tackle the above challenges, our motivation is that a succinct data sharing protocol should minimize bandwidth usage and not depend on the specific model architecture. Note that sharing model parameters in original FRL can be interpreted as sharing implicit client data summaries, since local model parameters capture the information from the input, which can be used to recover the raw data (Mothukuri et al., 2021). However, there is no guarantee that the information reflected by these parameters is sufficient or necessary to infer the global model. Instead, we can consider sharing concise, but sufficient client data summaries.

Sharing sufficient data summaries (also known as sufficient statistics (Jordan, 2009a)) of clients offers two benefits. 1) sufficient statistics maintain small data sizes, which reduces bandwidth usage. 2) sufficient statistics are model independent, which allows for heterogeneous model deployment. Based on this idea, we propose **Fed**erated Bayesian **Log**istic Regression (FedLog), a new FRL strategy. We consider the case that the body of the model is localized and the head is updated by the server. FedLog acquires sufficient statistics from the client data encoded by the body with an exponential family distribution. The sufficient statistics from each client are then sent to the server to determine the optimal head parameters by maximizing the posterior with Bayesian inference. The theoretical property of sufficient statistics ensures that sufficient information is captured with fixed size to infer global model parameters. Also, due to the Bayesian inference step, the number of communication rounds is reduced thus improving communication efficiency. Note that since the model body is not shared and the summation of sufficient statistics is non-invertible, FedLog also avoids potential privacy attacks through weight manipulation, GAN-based reconstruction, or large model memorization effects (Boenisch et al., 2021; Mothukuri et al., 2021). However, to ensure a formal privacy guarantee, we further incorporate the differential privacy framework to mitigate privacy leakage.

In summary, the paper makes the following contributions:

- FedLog: a new FRL algorithm, the model of which is carefully designed to provide a statistical interpretation for representation learning.

- Theoretical analyses of convergence and the shared messages being minimal sufficient statistics.

- Experiments demonstrating FedLog's low communication cost (as small as 0.09% of FedAvg) and fast convergence under multiple scenarios.

- Incorporation of DP and demonstration of a decent trade-off between privacy and utility.

## 2 Related Works

### 2.1 Federated Representation Learning

In FL, there is a collection of clients $c \in S$ wishing to collaborate. Each client holds their own data $\mathcal{D}_c = (\mathbf{X}_c, \mathbf{y}_c)$ locally. We seek to train some ML model with parameters $\boldsymbol{\theta}$ on these client data. Conventionally, we would centralize all the data $\mathcal{D} = \bigcup_{c \in S} \mathcal{D}_c$ and learn the model with $\mathcal{D}$. This approach becomes infeasible if the clients cannot share their data due to privacy concerns. FL intends to tackle this problem. A trusted central server is allowed to receive and send perturbed matrices that only contain limited information about raw data, such as model parameters.

One challenge of FL is how to aggregate model parameters learnt locally so that the resulting global consensus $\boldsymbol{\theta}^{t+1}$ is a better approximation to the centralized version than the last round $\boldsymbol{\theta}^t$ . This is especially challenging when the number of local training epochs $> 1$ (Karimireddy et al., 2020), due to non-linear loss functions and predictors. Thus, many FL algorithms simply resort to the averaging heuristic (McMahan et al., 2017; Liang et al., 2020; Arivazhagan et al., 2019; Collins et al., 2021; Achituve et al., 2021).

In FL, clients often have different data distributions $\Pr_c(\mathbf{X}), \Pr_c(\mathbf{y})$ or even $\Pr_c(\mathbf{y}|\mathbf{X})$. This is referred to as heterogeneous or non-i.i.d. clients. The averaging heuristic can drastically harm the global aggregation in terms of convergence rate and model utility with non-i.i.d. clients (Li et al., 2019). Thus, personalized FL (PFL) is introduced, where a global model is not mandatory, but each client could have their own model that best fits their data distribution (Tan et al., 2022a). Nevertheless, most algorithms still utilize the averaging heuristic for aggregation.

One line of FRL works approach PFL by localizing either the body (LG-FedAvg (Liang et al., 2020)) or the head (FedPer (Arivazhagan et al., 2019), FedRep (Collins et al., 2021)) of the client models, and shares the rest with the server for averaging. FedProto (Tan et al., 2022b) localizes the whole model, but averages feature representations by class and forces local models to learn similar representations. FedLog can also be interpreted as a representation learning algorithm, where we learn local representations by all the layers except the last one. However, unlike previous works that share model parameters and heuristically average them, we share sufficient statistics and update the global head with Bayesian inference.

Another line of works (CCVR (Luo et al., 2021), FedPFT (Beitollahi et al., 2024)) fit Gaussian Mixture Models (GMM) to local features extracted by a globally unified body, and share the natural parameters. The server then draws virtual features from the GMM and trains a global model. It is notable that these algorithms rely on pretrained FedAvg or foundation models to unify the body. Although FedLog also fits an exponential family distribution to local features, we do not need such pretrained models since our bodies are trained locally with any architecture. Also, we work with the canonical parameters and do not need to draw virtual samples from the learned distribution.

## 2.2 Other Related Works

**Communication efficient FL.**  Being orthogonal to FRL, communication efficient FL aims to directly decrease the communication overhead with optimization algorithms, client selection, and model compression (Wen et al., 2023). First, since local training epochs affect the rounds of global communication needed (McMahan et al., 2017), researchers proposed different local optimization methods to reduce the communication rounds (Liu et al., 2020; Wu & Wang, 2021; Wu et al., 2022). Second, some clients may contribute more to the global model, or are faster when uploading parameters. Thus, the global learning process can be accelerated by carefully selecting clients that meet these criteria (Liu et al., 2021b; Deng et al., 2021; Lai et al., 2021; Du et al., 2022). Additionally, the size of transmitted messages can be directly decreased by reducing or compressing the model parameters (Lu et al., 2020; Li & Xiao, 2021; Li et al., 2021). However, most such algorithms are efficient at the cost of model accuracy (Cai et al., 2022).

**Bayesian FL.**  Other Bayesian models have been explored to represent distributions over models and predictions in FL. The challenge is in the aggregation of the local posteriors. Various techniques have been proposed including personalized GPs (Achituve et al., 2021), posterior averaging (Al-Shedivat et al., 2020), online Laplace approximation (Liu et al., 2021a), Thompson sampling (Dai et al., 2020), MCMC sampling (Vono et al., 2022). These Bayesian FL techniques tend to emphasize calibration, approximating the posterior, or even different tasks. There is not much in common between them and our approach despite the use of Bayes theorem.

## 3 Method

### 3.1 Exponential Family

We start by introducing the definition of the exponential family.

**Definition 1.** Exponential family refers to a set of probability distributions of the following canonical form.

$$\Pr(\mathbf{x}|\boldsymbol{\eta}) = h(\mathbf{x})\exp(\boldsymbol{\eta}^\top \mathbf{T}(\mathbf{x}) - A(\boldsymbol{\eta})) \tag{1}$$

where $h(\mathbf{x}) : \mathbb{R}^p \to \mathbb{R}_{\geq 0}$, $\mathbf{T}(\mathbf{x}) : \mathbb{R}^p \to \mathbb{R}^d$, $A(\boldsymbol{\eta}) : \mathbb{R}^d \to \mathbb{R}$ are known functions.

Note that $A(\boldsymbol{\eta})$ is automatically determined by $h(\mathbf{x})$ and $\mathbf{T}(\mathbf{x})$, since it must normalize the probability density function (p.d.f.), so that the integral of the p.d.f. equals to 1: $A(\boldsymbol{\eta}) = \ln \int_{\mathbf{x}} h(\mathbf{x}) \exp(\boldsymbol{\eta}^\top T(\mathbf{x})) d\mathbf{x}$.

Many well-known distributions are included in the exponential family, such as Gaussian, binomial, Poisson, and Bernoulli distributions. We can always transform a distribution represented by natural parameters into its canonical form defined above. For example, a binomial distribution has the following p.d.f.:

$$\Pr(x|p) = \binom{n}{x} p^x (1-p)^{(n-x)}, x \in \{0, 1, \cdots, n\}$$

It can be rewritten as:

$$\Pr(x|\eta) = \binom{n}{x} \exp(\eta x - n \ln(1 + e^\eta)), \eta = \ln \frac{p}{1-p}$$

Exponential family has a few desirable properties, which makes it a perfect candidate for information sharing in PFL. 1) The **sufficient** statistic $\mathbf{T}(\mathbf{x})$ captures all the information of $\mathbf{x}$ that can be used to infer $\boldsymbol{\eta}$ (i.e., $\mathbf{x} \perp\!\!\!\perp \boldsymbol{\eta} | \mathbf{T}(\mathbf{x})$, conditional independence) (Jordan, 2009a). 2) If $\mathbf{x}_1, \mathbf{x}_2, \cdots, \mathbf{x}_n$ are i.i.d. samples from $\Pr(\mathbf{x}|\boldsymbol{\eta})$, the summation of sufficient statistics $\sum_{i=1}^n \mathbf{T}(\mathbf{x}_i)$ is a **complete** statistic for $\boldsymbol{\eta}$. It contains only information about $\boldsymbol{\eta}$, without ancillary information (Casella & Berger, 2015). 3) It is the only parametric distribution family with sufficient statistics of **fixed size** that does not grow with the sample size (Koopman, 1936). 4) It has known **conjugate priors** (Jordan, 2009b), a crucial property for Bayesian inference as we will elaborate later. Formal definitions of these properties are in Appendix B.

### 3.2 Model Construction

We detail our model construction and algorithm with the following notation.

**Notation.** Let $\tilde{\boldsymbol{\theta}}_c$ denote the parameters of a local neural network of client $c$. This network serves as a local body. Let $f_{\tilde{\boldsymbol{\theta}}_c} : \mathbb{R}^p \to \mathbb{R}^m$ be the function of the local neural network and $\boldsymbol{\Phi}_c = f_{\tilde{\boldsymbol{\theta}}_c}(\mathbf{X}_c)$ be the $m$ local features extracted from the local input. For convenience, we designate the first feature to be always 1. Let $n_{class}$ denote the total number of classes in a classification task, and $n_c$ denote the size of the local dataset. Let $\mathbf{e}_i \in \mathbb{R}^{n_{class}}$ be the standard basis (i.e. one-hot vector form) of label $i \in \{1, 2, \cdots, n_{class}\}$, and $\otimes : \mathbb{R}^m \times \mathbb{R}^{n_{class}} \to \mathbb{R}^{m*n_{class}}$ be the Kronecker product.

We assume the joint probability of each data point in $(\boldsymbol{\Phi}_c, \mathbf{y}_c)$, denoted as $\Pr(\boldsymbol{\phi}, y)$, is an exponential family with canonical parameters $\boldsymbol{\eta} \in \mathbb{R}^{m*n_{class}}$. We design the sufficient statistics $\mathbf{T}(\boldsymbol{\phi}, y)$ and the base measure $h(\boldsymbol{\phi}, y)$ to have the following form.

$$\mathbf{T}(\boldsymbol{\phi}, y) := \boldsymbol{\phi} \otimes \mathbf{e}_y \tag{2}$$

$$h(\boldsymbol{\phi}, y) := \frac{\exp\left(-\sum_{i=1}^m \phi_i^2\right)}{\sqrt{\pi^m}} \tag{3}$$

where $\phi_i \in \mathbb{R}$ denotes the $i^{th}$ entry of $\boldsymbol{\phi}$. One advantage of this specific instantiation of $\mathbf{T}$ and $h$ lies in the resulting conditional likelihood $\Pr(y|\boldsymbol{\phi}, \boldsymbol{\eta})$. This is in analogy to JEM (Grathwohl et al., 2019), an energy based model that has a similar $\mathbf{T}$. Let $\boldsymbol{\eta}_y \in \mathbb{R}^m$ denote the $((y-1)*m)^{th}$ to $(y*m)^{th}$ entries of $\boldsymbol{\eta}$, $y \in \{1, 2, \cdots, n_{class}\}$. Then,

$$\Pr(\boldsymbol{\phi}, y|\boldsymbol{\eta}) = \frac{\exp(\boldsymbol{\eta}_y^\top \boldsymbol{\phi} - \boldsymbol{\phi}^\top \boldsymbol{\phi} - A(\boldsymbol{\eta}))}{\sqrt{\pi^m}} \tag{4}$$

$$\Pr(y|\boldsymbol{\phi}, \boldsymbol{\eta}) = \frac{\Pr(\boldsymbol{\phi}, y|\boldsymbol{\eta})}{\Pr(\boldsymbol{\phi}|\boldsymbol{\eta})} = \frac{\Pr(\boldsymbol{\phi}, y|\boldsymbol{\eta})}{\sum_{y'=1}^{n_{class}} \Pr(\boldsymbol{\phi}, y'|\boldsymbol{\eta})} = \frac{\exp(\boldsymbol{\eta}_y^\top \boldsymbol{\phi})}{\sum_{y'=1}^{n_{class}} \exp(\boldsymbol{\eta}_{y'}^\top \boldsymbol{\phi})} \tag{5}$$

Eq. 5 is exactly the softmax function over $\boldsymbol{\eta}_y^\top \boldsymbol{\phi}$. This means we can take any deep neural network that extracts $m$ features, and append $\boldsymbol{\eta}$ as the last linear layer that maps the features to $n_{class}$ logits. Then, this composed neural network serves as a stand-alone classifier that computes the conditional likelihood $\Pr(y|\boldsymbol{\phi}, \boldsymbol{\eta})$.

However, Eq. 5 cannot be directly maximized at the server, since it requires knowledge of uncompressed representation-label pairs. Instead, we utilize Bayesian inference to optimize $\boldsymbol{\eta}$.

With Bayesian inference, $\boldsymbol{\eta}$ is treated as a random variable. A prior distribution $\Pr(\boldsymbol{\eta})$ can be specified to incorporate prior knowledge. Then, by Bayes' Theorem, the posterior distribution is:

$$\Pr(\boldsymbol{\eta}|\boldsymbol{\phi}, y) = \frac{\Pr(\boldsymbol{\phi}, y|\boldsymbol{\eta}) \Pr(\boldsymbol{\eta})}{\Pr(\boldsymbol{\phi}, y)} = \frac{\Pr(\boldsymbol{\phi}, y|\boldsymbol{\eta}) \Pr(\boldsymbol{\eta})}{\int_{\boldsymbol{\eta}} \Pr(\boldsymbol{\phi}, y|\boldsymbol{\eta}) \Pr(\boldsymbol{\eta}) d\boldsymbol{\eta}} \tag{6}$$

The integral in Eq. 6 and thus the posterior $\Pr(\boldsymbol{\eta}|\boldsymbol{\phi}, y)$ may not be tractable for arbitrary priors $\Pr(\boldsymbol{\eta})$. A convenient choice that guarantees analytical solutions is the conjugate prior. Given a likelihood $\Pr(\boldsymbol{\phi}, y|\boldsymbol{\eta})$, a prior is called its conjugate prior if $\Pr(\boldsymbol{\eta})$ and $\Pr(\boldsymbol{\phi}, y|\boldsymbol{\eta})$ follow the same distribution family. Specifically, if the likelihood is an exponential family, it has known conjugate priors (Jordan, 2009b).

$$\text{Prior:} \Pr(\boldsymbol{\eta}; \boldsymbol{\chi}, \nu) = f(\boldsymbol{\chi}, \nu) \exp(\boldsymbol{\eta}^\top \boldsymbol{\chi} - \nu A(\boldsymbol{\eta})) \tag{7}$$

$$\text{Posterior:} \Pr(\boldsymbol{\eta}|\boldsymbol{\phi}, y) = \Pr(\boldsymbol{\eta}; \boldsymbol{\chi} + \mathbf{T}(\boldsymbol{\phi}, y), \nu + 1) \tag{8}$$

where $\boldsymbol{\chi} \in \mathbb{R}^d, \nu \in \mathbb{R}$ are deterministic parameters of the prior, and $f(\boldsymbol{\chi}, \nu) : \mathbb{R}^d \times \mathbb{R} \to \mathbb{R}$ is automatically determined by $A(\boldsymbol{\eta})$: $f(\boldsymbol{\chi}, \nu)^{-1} = \int_{\boldsymbol{\eta}} \exp(\boldsymbol{\eta}^\top \boldsymbol{\chi} - \nu A(\boldsymbol{\eta})) d\boldsymbol{\eta}$.

Another advantage of our design is that $A$ has an explicit expression, which is intractable in JEM. Let $\boldsymbol{\eta}_{y,i} \in \mathbb{R}$ denote the $i^{th}$ entry of $\boldsymbol{\eta}_y$.

$$A(\boldsymbol{\eta}) = \ln \sum_{y=1}^{n_{class}} \int_{-\infty}^{\infty} \frac{\exp\left(\sum_{i=1}^{m} \boldsymbol{\eta}_{y,i} \phi_i - \phi_i^2\right)}{\sqrt{\pi^m}} d\boldsymbol{\phi} = \ln \sum_{y=1}^{n_{class}} \exp\left(\frac{\sum_{i=1}^{m} \eta_{y,i}^2}{4}\right) \tag{9}$$

Due to this analytical solution, we can directly optimize Eq. 8 without further approximations or sampling.

### 3.3 FedLog

Based on the above model, we propose our new algorithm FedLog, summarized in Alg. 1. At the beginning, the server initializes $\boldsymbol{\eta}$ (the global head) randomly. The clients initialize $\tilde{\boldsymbol{\theta}}_c$ (the local bodies) either completely randomly, or with the same random seed sent by the server to unify the initialization. Note $\tilde{\boldsymbol{\theta}}_c$ is not part of our exponential family assumption, thus we do not require them to have the same shape or architecture among different clients.

Parameters that we need to optimize are essentially $\tilde{\boldsymbol{\theta}}_c$ and $\boldsymbol{\eta}$, which can be done by maximizing $\Pr(y|\boldsymbol{\phi}, \boldsymbol{\eta})$ and $\Pr(\boldsymbol{\eta}|\boldsymbol{\phi}, y)$ in turns iteratively, similarly to the expectation-maximization algorithm. Concretely, all the clients $c \in S$ first fix the global head $\boldsymbol{\eta}$, and update their local bodies $\tilde{\boldsymbol{\theta}}_c$ with gradient descent. We derive the loss function as:

$$\mathcal{L}_c = -\sum_{i=1}^{n_c} \ln \Pr(\mathbf{y}_{c,i}|\boldsymbol{\Phi}_{c,i}, \boldsymbol{\eta}) = -\sum_{i=1}^{n_c} \ln \frac{\exp(\boldsymbol{\eta}_{y_{c,i}}^\top \boldsymbol{\Phi}_{c,i})}{\sum_{y=1}^{n_{class}} \exp(\boldsymbol{\eta}_y^\top \boldsymbol{\Phi}_{c,i})} \tag{10}$$

where $(\boldsymbol{\Phi}_{c,i}, \mathbf{y}_{c,i}) \in \mathbb{R}^m \times \mathbb{R}$ is the $i^{th}$ data point of $\boldsymbol{\Phi}_c, \mathbf{y}_c$. This is exactly the cross-entropy loss widely used in deep learning for classification tasks. Then, clients compute sufficient statistics of their local data $\sum_{i=1}^{n_c} \mathbf{T}(\boldsymbol{\Phi}_{c,i}, \mathbf{y}_{c,i}) = \sum_{i=1}^{n_c} \boldsymbol{\Phi}_{c,i} \otimes \mathbf{e}_{\mathbf{y}_{c,i}}$. They then send the summations and $n_c$ to the server for global head learning. As we have discussed with the introduction of the exponential family, the sufficient statistics contain all information in the representations that could be used to infer $\boldsymbol{\eta}$ in our model. The server only needs to know the summation of all the sufficient statistics, since

$$\Pr(\boldsymbol{\eta}|\boldsymbol{\Phi}_{c_1}, \mathbf{y}_{c_1}, \cdots, \boldsymbol{\Phi}_{c_k}, \mathbf{y}_{c_k}) = \Pr(\boldsymbol{\eta}; \boldsymbol{\chi} + \sum_{c \in S} \sum_{i=1}^{n_c} \boldsymbol{\Phi}_{c,i} \otimes \mathbf{e}_{\mathbf{y}_{c,i}}, \nu + \sum_{c \in S} n_c) \tag{11}$$

can be trivially inferred from Eq. 8. Note the size of the message sent by each client equals the size of the last linear layer. After receiving the sufficient statistics, the server computes $\boldsymbol{\Phi} = \sum_{c \in S} \sum_{i=1}^{n_c} \boldsymbol{\Phi}_{c,i} \otimes \mathbf{e}_{\mathbf{y}_{c,i}}, n =$

---

**Algorithm 1** FedLog ($\mathbf{X}_c, \mathbf{y}_c$: local data, $\tilde{\boldsymbol{\theta}}_c$ local body parameters, $\boldsymbol{\eta}$: global head parameters, $\boldsymbol{\chi}$: prior parameter, $\nu$: prior parameter, $\zeta$: local learning rate)

---

**for** each global update round **do**
    **Server:** sends $\boldsymbol{\eta}$ to clients
    **for** each **client** $c \in S$ **do**
        **for** each local update round **do**
            $\tilde{\boldsymbol{\theta}}_c \leftarrow \tilde{\boldsymbol{\theta}}_c - \zeta \nabla \mathcal{L}_c$ (Eq. 10)
        $\boldsymbol{\Phi}_c \leftarrow f_{\tilde{\boldsymbol{\theta}}_c}(\mathbf{X}_c)$
        Send $\sum_{i=1}^{n_c} \boldsymbol{\Phi}_{c,i} \otimes \mathbf{e}_{\mathbf{y}_{c,i}}, n_c$ to server
    **Server:**
    $\boldsymbol{\Phi} \leftarrow \sum_{c \in S} \sum_{i=1}^{n_c} \boldsymbol{\Phi}_{c,i} \otimes \mathbf{e}_{\mathbf{y}_{c,i}}, n \leftarrow \sum_{c \in S} n_c$
    $\boldsymbol{\eta} \leftarrow \arg\max_{\boldsymbol{\eta}} \Pr(\boldsymbol{\eta}; \boldsymbol{\chi} + \boldsymbol{\Phi}, \nu + n)$ (Eq. 12)

---

$\sum_{c \in S} n_c$ and updates the global head $\boldsymbol{\eta}$ by maximum a posteriori (MAP):

$$\boldsymbol{\eta} = \arg\max_{\boldsymbol{\eta}} \ln \Pr(\boldsymbol{\eta}; \boldsymbol{\chi} + \boldsymbol{\Phi}, \nu + n) = \arg\max_{\boldsymbol{\eta}} \ln \frac{\exp\left(\boldsymbol{\eta}^\top (\boldsymbol{\chi} + \boldsymbol{\Phi})\right)}{\left(\sum_{y=1}^{n_{class}} \exp(\boldsymbol{\eta}_y^\top \boldsymbol{\eta}_y / 4)\right)^{(\nu+n)}} \tag{12}$$

Note Eq. 12 is a convex optimization task. We can easily compute its sole maximum by gradient descent with complexity $O(m * n_{class})$. Then, the server sends $\boldsymbol{\eta}$ back to all the clients and starts the next round of updates. The process is repeated until convergence.

### 3.4 Interpretation and FedLog-C

In this section, we analyze our assumptions in more detail and give some insights about how FedLog works. The first assumption we made is that $\forall c \in S$, the local data points $\boldsymbol{\Phi}_c, \mathbf{y}_c$ are from the same exponential family distribution whose pdf is given by Eq. 4. In other words, we assume that the local bodies $\tilde{\boldsymbol{\theta}}_c$ transform their input, of any form, to the same representation space. This may seem infeasible at first glance since we do not directly aggregate the local body parameters, and they may follow any architecture. However, note we fix the global head $\boldsymbol{\eta}$ during local updates, by which the local bodies are forced to learn a universal representation space. See the synthetic experiment below for more details. This assumption allows principled discriminative training for the local bodies with cross-entropy loss, but unavoidably leaves a generative model for learning the global head. We can further see that $\Pr(\boldsymbol{\phi}|y, \boldsymbol{\eta}) \propto \exp(\boldsymbol{\eta}_y^\top \boldsymbol{\phi} - \boldsymbol{\phi}^\top \boldsymbol{\phi})$, which is the kernel of a multivariate Gaussian distribution. This means we essentially assumed a mixture of Gaussians for the local features $\boldsymbol{\phi}$. Although it seems similar to CCVR (Luo et al., 2021) and FedPFT (Beitollahi et al., 2024), which also fit GMM to local features, FedLog differs fundamentally from this line of work in both the problem setting and the methodology: i) CCVR and FedPFT assume that a pre-trained body is available, and every client uses the same body for feature extraction, while FedLog aims to train the whole model from scratch. For CCVR and FedPFT, local representations are not learned, but given for free; ii) CCVR and FedPFT simply sample additional data from the GMM, augment local datasets with these samples, and train the head in a non-Bayesian fashion, while FedLog constructs the head as canonical parameters and finds the posterior.

To mitigate the gap between the assumption and the actual feature distribution, we propose a variation: FedLog-C. An auxiliary loss is added during local training to encourage the local bodies to learn Gaussian-like clusters. Let $\boldsymbol{\Phi}_y$ denote the $((y-1)*m)^{th}$ to $(y*m)^{th}$ entries of $\boldsymbol{\Phi}$. Let $\boldsymbol{\Phi}_{y,0}$ denote the first entry of $\boldsymbol{\Phi}_y$. Then $\overline{\boldsymbol{\Phi}_y} = \boldsymbol{\Phi}_y / \boldsymbol{\Phi}_{y,0}$ is the global mean representation of class $y$. Inspired by contrastive learning (Schroff et al., 2015), we derive the new local loss as:

$$\mathcal{L}'_c = \mathcal{L}_c + \alpha \sum_{i=1}^{n_c} (\boldsymbol{\Phi}_{c,i} \otimes \mathbf{e}_{\mathbf{y}_{c,i}} - \overline{\boldsymbol{\Phi}_{\mathbf{y}_{c,i}}})^2 / n_c \tag{13}$$

where $\alpha \in \mathbb{R}_{\geq 0}$ controls how compact the clusters should be. Since clients need to know $\boldsymbol{\Phi}$, the server simply broadcasts the aggregated statistic to all the clients. Clients can optimize the same $\boldsymbol{\eta}$ locally, preserving the

same communication cost. This auxiliary loss does not directly enforce Gaussianity. It decreases variance and incentivizes unimodal and light-tailed clusters, which are more likely to be Gaussian since the Gaussian distribution is unimodal and light-tailed. As we will show empirically in Appendix D, this auxiliary loss successfully shapes the local representations to satisfy our Gaussian assumption.

The second assumption we made is the prior of the global head. Since $\boldsymbol{\eta}$ has support over $\mathbb{R}^{m*n_{class}}$, it is impossible to specify a uniform prior. Without any prior knowledge, we can set $\boldsymbol{\chi} = \mathbf{0}, \nu = 1$. The prior then becomes $\Pr(\boldsymbol{\eta}) \propto \exp(-A(\boldsymbol{\eta})) = (\sum_{y=1}^{n_{class}} \exp(\boldsymbol{\eta}_y^\top \boldsymbol{\eta}_y/4))^{-1}$. The p.d.f. takes its maximum at $\boldsymbol{\eta} = \mathbf{0}$ and decreases quickly as the norm of $\boldsymbol{\eta}_y$ grows larger. This is in analogy to the Lasso regularizer in the regression case, which prevents the model from learning coefficients with large absolute values due to noise or overfitting.

From the Bayesian view, FedLog can take any deep classifier, and make the last linear layer Bayesian. It essentially operates a Bayesian logistic regression model on the local representations. We start from a generative assumption and achieve the cross-entropy loss conditional likelihood usually assumed directly in Bayesian logistic regression. We obtained an analytical solution for the kernel of the posterior, which can be calculated easily by the summation of sufficient statistics. The shared statistic cannot be further compressed without losing information from the representations. We formalize this statement as the following theorem.

**Theorem 2.** *If $(\boldsymbol{\phi}_1, y_1), (\boldsymbol{\phi}_2, y_2), \cdots, (\boldsymbol{\phi}_n, y_n)$ are i.i.d. samples from the exponential family defined with Eq. 4, then $\mathbf{T}((\boldsymbol{\phi}_1, y_1), \cdots, (\boldsymbol{\phi}_n, y_n)) = \sum_{i=1}^{n} \boldsymbol{\phi}_i \otimes \mathbf{e}_{y_i}$ is a minimal sufficient statistic independent of every ancillary statistic.*

*Proof.* See Appendix A for the proof. □

From the federated representation learning view, FedLog has a one-layer global head and a deep local body. It iterates between learning local representations and learning global linear separators as if it has seen all the local representations. The two learning processes are completely separated, unlike the common paradigm where the local representations and the linear separators are often optimized jointly. The clients are only responsible for moving local representations to the correct sides of the fixed linear separator. The server is only responsible for finding the best linear separator given the local representations. As we will show with the experiments, we can converge faster than using the averaging heuristic.

### 3.5 Convergence Analysis

In this section, we analyze the convergence rate of FedLog in a non-convex setting. Similarly to Tan et al. (2022b) and Li et al. (2019), we make the following assumptions w.r.t. the local loss $\mathcal{L}_c$. Denote $\mathcal{L}_t(\mathcal{D}_c)$ as the local loss $\mathcal{L}_c$ at a specific timestep $t$. We will always omit $(\mathcal{D}_c)$ in $\mathcal{L}_t(\mathcal{D}_c)$ for simplicity.

**Assumption 1** (Lipschitz Smooth). Each local objective function $\mathcal{L}$ is $L_1$-Lipschitz smooth w.r.t. the body $\boldsymbol{\theta}$, so the gradient of any local objective function is $L_1$-Lipschitz continuous.

$$||\nabla_{\boldsymbol{\theta}_c}\mathcal{L}_{t_1} - \nabla_{\boldsymbol{\theta}_c}\mathcal{L}_{t_2}||_2 \leq L_1||\boldsymbol{\theta}_{t_1,c} - \boldsymbol{\theta}_{t_2,c}||_2^2, \forall t_1, t_2 > 0, \forall c \in S$$

, implying

$$\mathcal{L}_{t_1} - \mathcal{L}_{t_2} \leq \langle \nabla_{\boldsymbol{\theta}_c}\mathcal{L}_{t_2}, \boldsymbol{\theta}_{t_1,c} - \boldsymbol{\theta}_{t_2,c} \rangle + \frac{L_1}{2}||\boldsymbol{\theta}_{t_1,c} - \boldsymbol{\theta}_{t_2,c}||_2^2$$

**Assumption 2** (Unbiased Gradient and Bounded Variance). Let $\xi_c \sim \mathcal{D}_c$ be a random batch of local datasets $\mathcal{D}_c$. The stochastic gradient $g_{t,c} = \nabla_{\boldsymbol{\theta}_c}\mathcal{L}(\xi_{t,c})$ is an unbiased estimator of the local gradient for each client.

$$\mathbb{E}_{\xi_c \sim \mathcal{D}_c}[g_{t,c}] = \nabla_{\boldsymbol{\theta}_c}\mathcal{L}_t$$

Its variance is bounded by $\sigma^2$.

$$\mathbb{E}[||g_{t,c} - \nabla_{\boldsymbol{\theta}_c}\mathcal{L}_t||_2^2] \leq \sigma^2$$

**Assumption 3** (Lipschitz Smooth). Each local objective function $\mathcal{L}$ is $L_2$-Lipschitz smooth w.r.t. the head $\boldsymbol{\eta}$.

$$||\nabla_{\boldsymbol{\eta}}\mathcal{L}_{t_1} - \nabla_{\boldsymbol{\eta}}\mathcal{L}_{t_2}||_2 \leq L_2||\boldsymbol{\eta}_{t_1} - \boldsymbol{\eta}_{t_2}||_2^2, \forall t_1, t_2 > 0$$

$$\mathcal{L}_{t_1} - \mathcal{L}_{t_2} \leq \langle \nabla_{\boldsymbol{\eta}}\mathcal{L}_{t_2}, \boldsymbol{\eta}_{t_1} - \boldsymbol{\eta}_{t_2} \rangle + \frac{L_2}{2}||\boldsymbol{\eta}_{t_1} - \boldsymbol{\eta}_{t_2}||_2^2$$

**Assumption 4** (Heterogeneity). Let $\mathcal{D} = \cup_{c \in S} \mathcal{D}_c$ be the union of all datasets. For each client $c$, the angle between local gradients and global gradients w.r.t. the head $\boldsymbol{\eta}$ is bounded:

$$\exists H > 0, \langle \nabla_{\boldsymbol{\eta}} \mathcal{L}_t, \nabla_{\boldsymbol{\eta}} \mathcal{L}_t(\mathcal{D}) \rangle \geq H ||\nabla_{\boldsymbol{\eta}} \mathcal{L}_t||_2 ||\nabla_{\boldsymbol{\eta}} \mathcal{L}_t(\mathcal{D})||_2, \forall t > 0$$

We also assume the following norm is bounded:

$$\exists F > 0, ||\nabla_{\boldsymbol{\eta}} \mathcal{L}_t||_2 \geq F ||\nabla_{\boldsymbol{\eta}} \mathcal{L}_t(\mathcal{D})||_2, \forall t > 0$$

Assumption 4 essentially bounds the deviation between $\nabla_{\boldsymbol{\eta}} \mathcal{L}_t$ and $\nabla_{\boldsymbol{\eta}} \mathcal{L}_t(\mathcal{D})$. Since local representations can be unified by fixing $\boldsymbol{\eta}$ during local training, we expect $\nabla_{\boldsymbol{\eta}} \mathcal{L}_t$ and $\nabla_{\boldsymbol{\eta}} \mathcal{L}_t(\mathcal{D})$ to be at worst perpendicular. Also, since we never update $\boldsymbol{\eta}$ locally, we expect it not to fall into any local stationary point, and $||\nabla_{\boldsymbol{\eta}} \mathcal{L}_t||_2 > 0$ should be bounded from below.

In the following results, global communication rounds are denoted as $t \in \{0, \cdots, T\}$, local iterations as $e \in \{1, 2, \cdots, E\}$, and iterations of optimizing the global head as $k \in \{0, 1, 2, \cdots, K\}$. A local iteration step is denoted as $t(E + K) + e$, and a global gradient descent step is denoted as $t(E + K) + E + k$. With such assumptions, we prove:

**Theorem 3** (One-round deviation). *Let Assumptions 1-4 hold. For any client, after each communication round, we have,*

$$\mathbb{E}[\mathcal{L}_{(t+1)(E+K)}] \leq \mathcal{L}_{t(E+K)} - (\zeta - \frac{L_1 \zeta^2}{2}) \sum_{e=0}^{E-1} ||\nabla_{\boldsymbol{\theta}} \mathcal{L}_{t(E+K)+e}||_2^2 + \frac{L_1 E \zeta^2 \sigma^2}{2}$$

$$- (\beta K H F - \frac{L_2 \beta^2}{2}) \sum_{k=0}^{K-1} ||\nabla_{\boldsymbol{\eta}} \mathcal{L}_{t(E+K)+E+k}(\mathcal{D})||_2^2$$

Thm. 3 bounds the deviation in the local loss of any class after one communication round. We can choose $\zeta$ and $\beta$, the learning rates of body and head optimization, to guarantee convergence.

**Theorem 4** (Guaranteed convergence). *The expectation of the loss function $\mathbb{E}[\mathcal{L}]$ of any client monotonically decreases in every communication round when*

$$\zeta_{e'} < \frac{2(\sum_{e=0}^{e'} ||\nabla_{\boldsymbol{\eta}} \mathcal{L}_{t(E+K)+e}||_2^2)}{L_1 (\sum_{e=0}^{e'} ||\nabla_{\boldsymbol{\eta}} \mathcal{L}_{t(E+K)+e}||_2^2 + E\sigma^2)}, e' \in \{0, 1, \cdots, E-1\}, \beta < \frac{2KHF}{L_2}$$

Thm 4 makes sure FedLog converges since the expectation of the local loss monotonically decreases. We further provide the convergence rate:

**Theorem 5** (Convergence rate). *Denote $\Delta = \mathcal{L}_0 - \mathcal{L}^*$, where $\mathcal{L}^*$ is the minimum value of $\mathcal{L}$. If $\zeta < min(\frac{2}{L_1}, \frac{2K\epsilon}{L_1(K+\sigma^2)}), \beta < \frac{2KHF}{L_2}$, denote $C_1 = \frac{2\beta KHF - L_2 \beta^2}{2\zeta - L_1 \zeta^2} > 0$; then for any client, given any $\epsilon > 0$, after*

$$T > \frac{2\Delta}{E\zeta(\epsilon K(2 - L_1 \zeta) - L_1 \zeta \sigma^2)}$$

*communication rounds*

$$\frac{1}{TEK} \sum_{t=0}^{T-1} \sum_{e=0}^{E-1} \mathbb{E}[||\nabla_{\boldsymbol{\theta}} \mathcal{L}_{t(E+K)+e}||_2^2] + \frac{C_1}{TEK} \sum_{t=0}^{T-1} \sum_{k=0}^{K-1} \mathbb{E}[||\nabla_{\boldsymbol{\eta}} \mathcal{L}_{t(E+K)+E+k}(\mathcal{D})||_2^2] < \epsilon$$

Thm. 5 shows that after $T$ communication rounds, the mean gradient of any local loss $\frac{1}{TE} \sum_{t=0}^{T-1} \sum_{e=0}^{E-1} \mathbb{E}[||\nabla_{\boldsymbol{\theta}} \mathcal{L}_{t(E+K)+e}||_2^2]$ and any global loss $\frac{1}{TK} \sum_{t=0}^{T-1} \sum_{k=0}^{K-1} \mathbb{E}[||\nabla_{\boldsymbol{\eta}} \mathcal{L}_{t(E+K)+E+k}(\mathcal{D})||_2^2]$ can decrease to arbitrarily small $\epsilon > 0$. Full proofs of all the presented theorems are in Appendix B.

### 3.6 Differential Privacy

FL can be combined with formal mechanisms such as differential privacy (DP) (Wei et al., 2020; Triastcyn & Faltings, 2019) or secure multi-party computation (MPC) (Truex et al., 2019; Byrd & Polychroniadou, 2020; Li et al., 2020), to provide formal privacy guarantees. We now extend FedLog to be differentially private.

$(\epsilon, \delta)$-DP protects clients' privacy by adding noise to the shared information so that the adversaries cannot effectively tell if any record is included in the dataset (controlled by $\epsilon > 0$) at most times (controlled by $0 \leq \delta < 1$) (Kerkouche et al., 2021).

**Definition 6.** A mechanism $M_{DP}$ satisfies $(\epsilon, \delta)$-DP if for any two datasets $\mathcal{D}, \mathcal{D}'$ that differ by only one record (i.e. $|(\mathcal{D} - \mathcal{D}') \cup (\mathcal{D}' - \mathcal{D})| = 1$), and for any possible output $O \in Range(M_{DP})$,

$$\Pr_{O \sim M_{DP}(\mathcal{D})} \left[ \log \left( \frac{\Pr[M_{DP}(\mathcal{D}) = O]}{\Pr[M_{DP}(\mathcal{D}') = O]} \right) > \epsilon \right] < \delta$$

Intuitively, $(\epsilon, \delta)$-DP guarantees that the inner log ratio, considered as the information loss leaked to the adversaries, is bounded by the privacy budget $\epsilon$ with probability $\delta$. We add Gaussian noise to shared sufficient statistics as follows.

**Theorem 7.** *If the absolute value of features is clipped to $b$ and there are in total $k$ global update rounds, FedLog messages satisfy $(\epsilon, \delta)$-DP with additive Gaussian noise $\mathbf{T}'(\mathbf{\Phi}_c, \mathbf{y}_c) := \mathbf{T}(\mathbf{\Phi}_c, \mathbf{y}_c) + \mathcal{N}(\mathbf{0}, \sigma^2 \mathbf{I})$, where $\sigma = \sqrt{8k(1 + (m-1) * b^2) \ln(e + \epsilon/\delta)}/\epsilon$.*

*Proof.*

$$\max_{\mathcal{D}, \mathcal{D}'} ||\mathbf{T}(\mathcal{D}) - \mathbf{T}(\mathcal{D}')||_2 = \max_{\phi, y} ||\mathbf{T}(\phi, y)||_2 = \sqrt{1 + (m-1) * b^2}$$

See more details in Appendix C. $\qquad \square$

When $n$ is large enough, the amount of noise (independent of $n$) becomes negligible to the model. Optionally, with secure MPC or central DP, it is sufficient to add such Gaussian noise once globally to $\mathbf{\Phi}$ each global update round. The server then computes the posterior normally with the noisy sufficient statistics.

## 4 Experiments

### 4.1 Synthetic

We designed the following synthetic experiment to justify our claim that FedLog can learn universal local representation spaces without sharing local bodies, and argue why FedLog can converge faster than prior arts. As shown in the top left image of Fig. 1, we first sample 80 two-dimensional training data points $(x_1, x_2)$ uniformly from the $[-5, 5] \times [-5, 5]$ square. Data points are separated into class 0 (blue) and class 1 (green) by the circle located at the origin and of radius $26/7$. These data points are further divided evenly into two sets, client 0 (circles) and client 1 (triangles), based on ordered $x_1$ to simulate non-i.i.d. clients. Client 0 operates a three-layer fully connected body, while client 1 operates a two-layer fully connected body, to simulate clients with different computational resources and showcase the flexibility. We compare FedLog to FedProto and LG-FedAvg with a one-layer global head, in which case the size of shared messages is the same among all three algorithms. The top middle image of Fig. 1 shows the local representations and linear separators with random initialization. Lines are the linear separators induced by the global head. To be fair, the heads of both clients are initialized to be the same for LG-FedAvg and FedProto. We run FedLog, LG-FedAvg, and FedProto for one global update round, with 1 to 30 local iterations. The top right, bottom left, and bottom middle images of Fig. 1 respectively show the models FedLog, LG-FedAvg, and FedProto converge to locally. LG-FedAvg learns very different local representations and linear separators even if the last layer is initialized to be the same, and the averaged global linear separator (red line) is clearly suboptimal. This is because it jointly updates the feature extractor (body) and the linear separator (head), and the local updates diverge in different directions. On the contrary, FedLog clients learn universal local representations with the fixed last

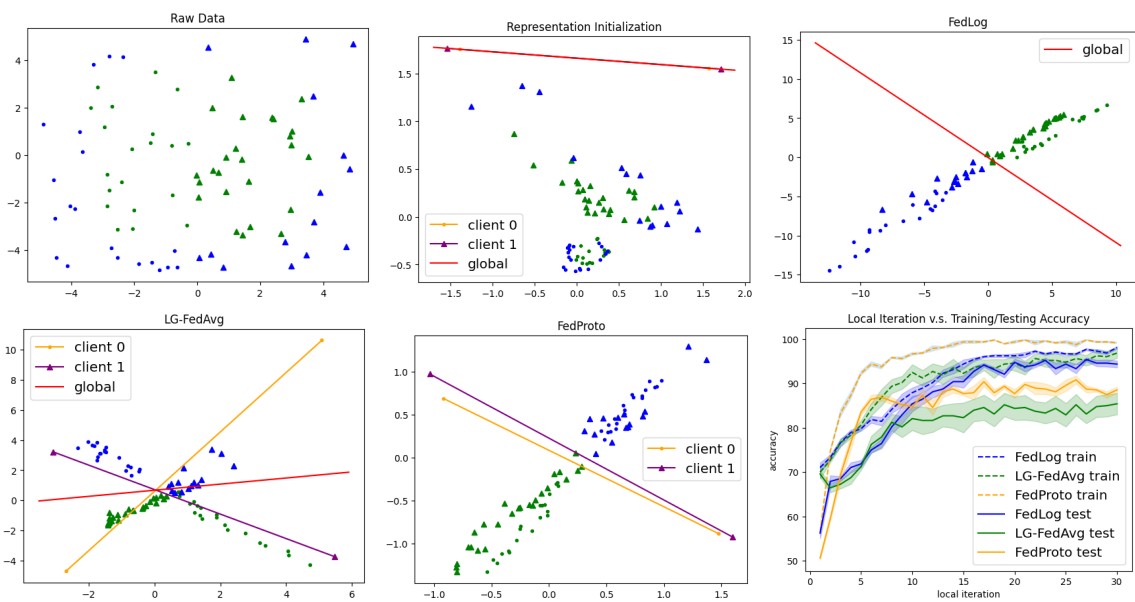

Figure 1: Synthetic experiments. Blue and green dots are local representations of class 0 and class 1. Circles and triangles show data distributed to client 0 or client 1. Lines are linear separators. Accuracy results are averaged over 6 seeds, and colored area shows mean ± std.

linear layer, and the server is able to draw the linear separator as if it has seen all client data. FedProto can learn similar local representations since it localizes the whole model and forces local models to learn similar representations to the globally averaged feature prototypes. information through the prototypes only. However, this extreme model construction gives up the generalization ability, making it prune to over-fitting with multiple local iterations. Finally, the bottom right image shows the training and testing accuracy v.s. local iterations. The testing data points are sampled i.i.d. from the same distribution. The results show: i) FedLog makes more progress in one global update round than other averaging-based prior arts; ii) FedLog is resistant to overfitting, as the difference between training and testing accuracy is small; iii) FedLog can learn universal local representations by fixing the last layer, even with different initialization and architectures of the body.

## 4.2 Communication Cost

To show FedLog achieves better accuracy with less communication with non-i.i.d. clients, we conduct experiments on MNIST, CIFAR10, and CIFAR100. We compare FedLog and FedLog-C with the following baselines: i) FedAvg (McMahan et al., 2017), which averages the whole model each global update round; ii) LG-FedAvg (Liang et al., 2020), which localizes bodies and averages heads. Two variants are reported: LG-FedAvg 1 which maintains one global layer, and LG-FedAvg 2 which maintains two global layers; iii) FedPer (Arivazhagan et al., 2019), which localizes heads and averages bodies; iv) FedRep (Collins et al., 2021), which also localizes heads and averages bodies, but trains local heads and bodies separately. FedRep 1 maintains one local layer as reported in the original paper, and FedRep 2 maintains two local layers for better communication efficiency; v) CS-FL (Li et al., 2021), which compresses shared messages with the compressed sensing framework; vi) FedBabu (Oh et al., 2021), which averages bodies and never updates heads; vii) FedProto (Tan et al., 2022b), which averages local feature representations by class and forces local models to learn similar representations; viii) FedDBE (Zhang et al., 2024), which averages both bodies and heads but accelerates convergence by learning a domain representation bias; ix) LG-FedAlt, which freezes the head of LG-FedAvg 1 during local training and is an instance of FedAlt (Pillutla et al., 2022). Convolutional neural networks of the same architecture and initialization are used for all the algorithms. FedPer localizes the last two layers.

Table 1: Testing accuracy and communication cost reported for MNIST, CIFAR10, and CIFAR100. Accuracy reports the mean ± standard error of the testing accuracy over 10 seeds. Higher is better. Communication cost reports the total message size transmitted between clients and the server. Lower is better. ⇑ denotes significantly higher results with $p < 0.01$; ⇓ denotes significantly lower results with $p < 0.01$.

| | MNIST | | CIFAR10 | | CIFAR100 | |
|---|---|---|---|---|---|---|
| | accuracy | comm cost | accuracy | comm cost | accuracy | comm cost |
| FedAvg | 89.76±0.69⇓ | 3.45±0.02Gb⇑ | 26.29±0.44⇓ | 37.9±1.31Gb⇑ | 13.34±0.15⇓ | 69.1±0.47Gb⇑ |
| LG-FedAvg 1 | 97.85±0.05⇓ | 4.81±0.31Mb⇑ | 86.57±0.29⇓ | 0.26±0.02Gb⇑ | 55.00±0.26⇓ | 4.34±0.18Gb⇑ |
| LG-FedAvg 2 | 98.18±0.06⇓ | 0.16±0.07Gb⇑ | 85.56±0.32⇓ | 3.38±0.32Gb⇑ | 54.90±0.24⇓ | 9.53±0.53Gb⇑ |
| FedPer | 96.16±0.19⇓ | 0.65±0.05Gb⇑ | 83.54±0.40⇓ | 27.7±1.50Gb⇑ | 52.82±0.21⇓ | 42.7±1.47Gb⇑ |
| FedRep 1 | 98.15±0.07⇓ | 0.18±0.02Gb⇑ | 83.23±0.40⇓ | 27.34±2.72Gb⇑ | 54.92±0.25⇓ | 32.59±1.84Gb⇑ |
| FedRep 2 | 95.51±0.29⇓ | 36.3±3.38Mb⇑ | 82.96±0.35⇓ | 15.3±1.33Gb⇑ | 48.70±0.29⇓ | 11.0±0.40Gb⇑ |
| CS-FL | 79.65±1.22⇓ | 0.35±0.01Gb⇑ | 23.60±1.08⇓ | 2.72±0.41Gb⇑ | 4.52±0.15⇓ | 13.7±0.22Gb⇑ |
| FedBabu | 86.30±1.04⇓ | 2.25±0.02Gb⇑ | 25.37±0.44⇓ | 34.7±2.29Gb⇑ | 9.70±0.16⇓ | 59.3±0.31Gb⇑ |
| FedProto | 98.19±0.06⇓ | **3.02±0.17Mb** | 87.37±0.26⇓ | 0.18±0.01Gb⇑ | 55.32±0.19⇓ | 3.01±0.12Gb⇑ |
| FedDBE | 96.79±0.34⇓ | 1.71±0.39Gb⇑ | 72.77±0.79⇓ | 38.3±0.95Gb⇑ | 36.67±0.85⇓ | 55.1±2.80Gb⇑ |
| LG-FedAlt | 98.07±0.05⇓ | 3.91±0.27Mb⇑ | 86.54±0.3⇓ | 0.25±0.02Gb⇑ | 54.74±0.17⇓ | 4.47±0.16Gb⇑ |
| **FedLog** | 98.15±0.05⇓ | **3.18±0.31Mb** | 87.08±0.22⇓ | 0.14±0.01Gb⇑ | 56.46±0.27⇓ | **2.38±0.09Gb**⇓ |
| **FedLog-C** | **98.41±0.07** | **3.18±0.15Mb** | **87.57±0.25** | **0.11±0.01Gb** | **56.78±0.26** | 2.74±0.12Gb |

We first distribute the training set into $(50, 100, 100)$ heterogeneous clients for (MNIST, CIFAR10, CIFAR100) respectively. Each client takes only $(2, 2, 10)$ classes. The testing set is distributed similarly to the clients, following the same distribution as the training data. Specially, for **MNIST only**, to simulate the difficult situation where clients do not have sufficient local data, we train on 5% of the training set, while testing on the whole testing set. All the clients start from the same initialization, and all the algorithms are run for $(100, 100, 150)$ global update rounds. The testing accuracy is recorded after each round. Hyperparameters are optimized beforehand through grid search. We report results of different levels of heterogeneity in Appendix D. See Appendix E for the model architectures, hyperparameters used by each algorithm, and other details.

We report mean ± standard error of the testing accuracy resulting from 10 seeds in Table 1. We measured the statistical significance of the results compared to FedLog-C with one-tailed Wilcoxon signed-rank tests (Wilcoxon, 1992). Similarly, we report the total communication cost, namely the total size of messages transmitted between clients and the server, to reach a reasonable accuracy threshold (97%, 83%, and 53% respectively). If it is never reached, we stop counting at the round with the highest accuracy. These thresholds are the largest integers of accuracy which all competitive algorithms (FedLog, LG-FedAvg, FedProto) have reached in 10 seeds.

The results show FedLog can achieve statistically significant accuracy improvement compared to prior arts, with the least communication cost as small as 0.09%, 0.29%, and 3.44% of the communication cost of FedAvg for MNIST, CIFAR10, and CIFAR100 respectively. The faster convergence observed with FedLog is consistent with the synthetic experiment, where fixing the global head prevents local representation drift and allows each communication round to make more effective global progress. The closest baseline is FedProto, which has a slightly lower performance and requires more communication on CIFAR10 and CIFAR100 because it localizes the whole model and uses FedAvg instead of Bayesian inference for aggregation.

## 4.3 Large Datasets

We compare FedLog and FedLog-C to LG-FedAvg and FedProto on the large datasets Celeba (Liu et al., 2015) and Sent140 (Go et al., 2009) preprocessed by LEAF (Caldas et al., 2018). To test how these algorithms perform in situations where only a few communication rounds are available, we run each algorithm to optimum locally before global aggregation.

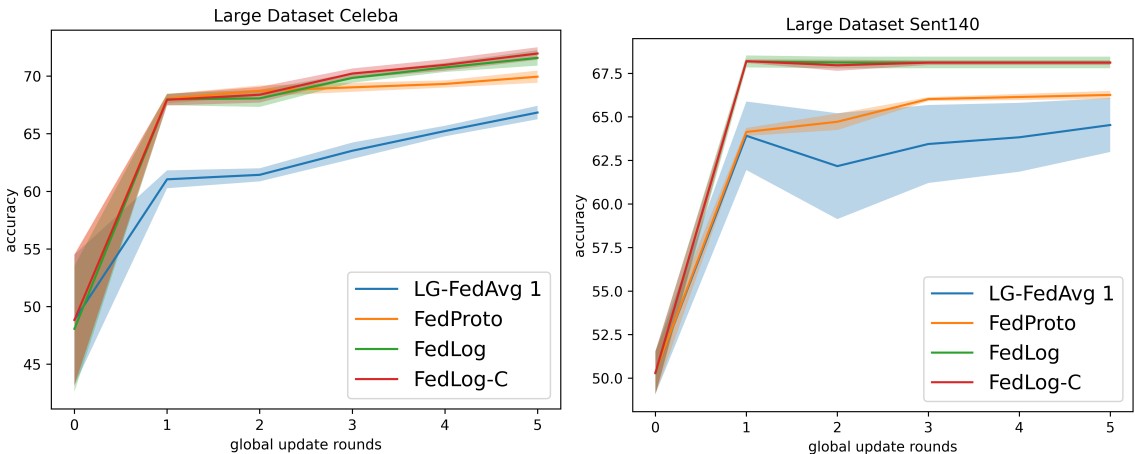

Figure 2: Left: Celeba. Right: Sent140. Mean client accuracy over global communication rounds is reported. Colored area shows mean $\pm$ std.

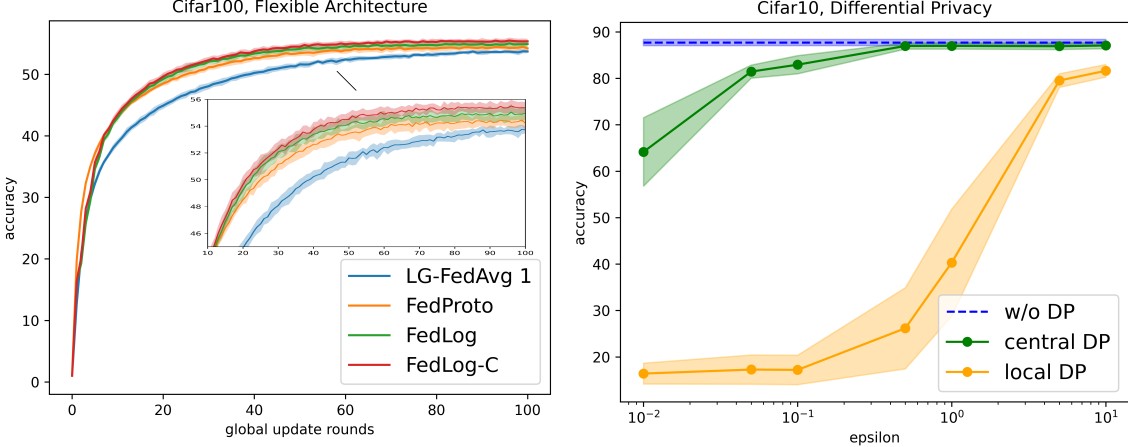

Figure 3: Left: flexible architecture. Right: differential privacy. Colored area shows mean $\pm$ standard error.

**Celeba.** We sampled 70838 images from 2360 clients. Each client represents a celebrity, and the local data comprise images of the same person. The task is to classify if the celebrity is smiling. MobileNetV2 (Sandler et al., 2018) is implemented as the classifier model for all algorithms. The results are shown in the left graph of Fig. 2. FedLog-C performs statistically significantly better than FedProto (**71.96$\pm$0.18** v.s. 69.98$\pm$0.16$^{\Downarrow}$, $p = 0.001$), LG-FedAvg (66.84$\pm$0.19$^{\Downarrow}$, $p = 0.001$), and FedLog (71.19$\pm$0.45$^{\downarrow}$, $p = 0.05$).

**Sent140.** We removed any client with less than 5 data points, and sampled 307413 data points from 33620 clients. Each client represents a Twitter user, and the local data are English posts. The task is sentiment analysis. A Transformer (Vaswani et al., 2017) encoder is implemented as the classifier model for all algorithms. Since few data is available for each client, we randomly select 10% clients to be the validation set. If the validation accuracy stops improving, the training process is considered terminated. The results are shown in the right graph of Fig. 2. FedLog and FedLog-C clearly converge faster and FedLog-C performs statistically significantly better than FedProto (**68.20$\pm$0.03** v.s. 66.27$\pm$0.08$^{\Downarrow}$, $p = 0.001$) and LG-FedAvg (64.62$\pm$0.49$^{\Downarrow}$, $p = 0.001$). The difference between FedLog-C and FedLog is not significant (68.18$\pm$0.11, $p = 0.34$), but FedLog-C is more stable as it has a smaller standard error.

Both datasets exhibit strong client heterogeneity, and fixing the global head enables the server to aggregate client information coherently, even when local data distributions differ significantly, which explains the improved performance here.

### 4.4   Flexible Architecture

We simulate the situation where clients have different computational resources on CIFAR100. We randomly select half of the clients and assign them a smaller convolutional neural network, where the second last fully connected layer is removed. Different clients start from different local body initializations, but the global head is unified. We compare FedLog and FedLog-C to LG-FedAvg 1 and FedProto. As shown in the left graph of Fig. 3, FedLog-C and FedLog converge faster than LG-FedAvg and FedProto. The accuracy of FedLog-C is also statistically significantly higher than FedProto ($\mathbf{55.86 \pm 0.11}$ v.s. $54.74 \pm 0.13^{\Downarrow}$, $p = 0.001$), LG-FedAvg ($54.03 \pm 0.08^{\Downarrow}$, $p = 0.001$), and FedLog ($55.31 \pm 0.15^{\Downarrow}$, $p = 0.001$). In this experiment, clients with different body architectures still learn compatible local representations due to the shared fixed head, demonstrating that FedLog naturally supports architectural flexibility without sacrificing convergence.

### 4.5   Differential Privacy

We conduct experiments to show the trade-off between privacy budget $\epsilon$ and the accuracy of FedLog on CIFAR10. We add an activation function to clip the extracted features to $b = 2$. Following a common practice in FL (Wei et al., 2020), we set $\delta = 0.01$. As shown in the right graph of Fig. 3, the accuracy of differentially private FedLog quickly grows back to optimum when $\epsilon \geq 0.5$, if the server is trusted or MPC is implemented so central DP is applicable. This is a strong privacy budget that shows FedLog performs well without sacrificing clients' privacy. Otherwise, local DP can have more impact on the model utility, but the accuracy is still acceptable when $\epsilon \geq 5.0$.

## 5   Limitations

One limitation of FedLog is that the algorithm works only for classification. Although it can be simply extended to regression with a Bayesian linear regression head and the idea of sharing sufficient statistics, the corresponding message size is not linear in the number of features $m$ anymore ($\mathbf{\Phi}^{\top}\mathbf{\Phi}$ and $\mathbf{\Phi y}$, $O(m^2 + m)$). Thus, it is not as communication efficient as the classification head. FedLog also assumes an exponential family distribution with a prior of the form $exp(-A(\boldsymbol{\eta}))$ and local transformed features distributed according to a mixture of Gaussians. However, the mixture of Gaussian assumption is mitigated in FedLog-C by introducing an auxiliary loss that helps satisfy the assumption. A benefit of the exponential family and mixture of Gaussian assumptions is that the data summaries are provably sufficient statistics (Thm. 2) and we obtain a closed-form solution for the normalization constant (Eq. 9).

## 6   Conclusion

We proposed FedLog that shares local data summaries instead of model parameters. FedLog assumes an exponential family model on local representations and learns a global linear separator with the summation of sufficient statistics. FedLog can learn universal local representations without sharing the bodies. Experiments show statistically significant improvements compared to prior arts, with the least communication cost. It is also effective with flexible architectures and formal DP guarantees. Based on experimental results, FedLog-C consistently outperforms FedLog in most settings. However, the results can be sensitive to the extra hyperparameter $\alpha$ introduced in FedLog-C. Thus, if hyperparameter searching is not feasible or differential privacy is needed, FedLog can be used instead. Otherwise, FedLog-C provides better accuracy and less communication cost.

### Acknowledgments

Resources used in this work were provided by Huawei Canada, the Province of Ontario, the Government of Canada through CIFAR, companies sponsoring the Vector Institute (`https://vectorinstitute.ai/about/current-partners/`), the Natural Sciences and Engineering Council of Canada and a grant from IITP & MSIT of Korea (No. RS-2024-00457882, AI Research Hub Project).

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

## A  Sufficiency

This section discusses formal definitions of sufficient and other statistics, based on Chapter 6 of Casella & Berger (2015).

**Sufficient Statistics.**

**Definition A.8** (6.2.1 in Casella & Berger (2015)). A statistic $\mathbf{T}(\mathbf{x})$ is a **sufficient** statistic for $\boldsymbol{\eta}$ if the conditional distribution of the sample $\mathbf{x}$ given the value of $\mathbf{T}(\mathbf{x})$ does not depend on $\boldsymbol{\eta}$.

Intuitively, a sufficient statistic captures all the information about $\boldsymbol{\eta}$ in $\mathbf{X}$. In the case of the exponential family, the following theorem applies:

**Theorem A.9** (6.2.10 in Casella & Berger (2015)). *Let $\mathbf{x}_1, \cdots, \mathbf{x}_n$ be i.i.d. observations from an exponential family whose p.d.f. is given by $\Pr(\mathbf{x}|\boldsymbol{\eta}) = h(\mathbf{x})\exp(\boldsymbol{\eta}^\top \mathbf{T}(\mathbf{x}) - A(\boldsymbol{\eta}))$, then $\mathbf{T}(\mathbf{x}_1, \cdots, \mathbf{x}_n) = \sum_{i=1}^n \mathbf{T}(\mathbf{x}_i)$ is a sufficient statistic for $\boldsymbol{\eta}$.*

**Minimal Sufficient Statistics.**

**Definition A.10** (6.2.11 in Casella & Berger (2015)). A sufficient statistic $\mathbf{T}(\mathbf{x})$ is **minimal** if, for any other sufficient statistic $\mathbf{T}'(\mathbf{x})$, $\exists$ a function $h$ such that $h(\mathbf{T}'(\mathbf{x})) = \mathbf{T}(\mathbf{x})$.

A minimal sufficient statistic achieves the greatest possible data reduction for a sufficient statistic. Whether a sufficient statistic is minimal can be verified by the following theorem.

**Theorem A.11** (6.2.13 in Casella & Berger (2015)). *If for all sample points $\mathbf{x}, \mathbf{x}'$ from the distribution with p.d.f. $\Pr(\mathbf{x}|\boldsymbol{\eta})$, $\frac{\Pr(\mathbf{x}|\boldsymbol{\eta})}{\Pr(\mathbf{x}'|\boldsymbol{\eta})}$ is independent of $\boldsymbol{\eta}$ iff $\mathbf{T}(\mathbf{x}) = \mathbf{T}(\mathbf{x}')$, then $\mathbf{T}(\mathbf{x})$ is minimal sufficient.*

**Ancillary Statistics.**

**Definition A.12** (6.2.16 in Casella & Berger (2015)). A statistic $\mathbf{S}(\mathbf{x})$ is an **ancillary** statistic if it is independent of the parameters $\boldsymbol{\eta}$.

As shown by the definition, an ancillary statistic on its own contains no information about the parameters $\boldsymbol{\eta}$.

**Complete Statistics.**

**Definition A.13** (6.2.21 in Casella & Berger (2015)). A family of distributions is called **complete** if $\mathbb{E}_{\boldsymbol{\eta}} g(\mathbf{T}) = 0, \forall \boldsymbol{\eta}$ implies $\Pr_{\boldsymbol{\eta}}(g(\mathbf{T}) = 0) = 1, \forall \boldsymbol{\eta}$. Then $\mathbf{T}(\mathbf{x})$ is a **complete** statistic.

This definition is less intuitive and harder to interpret. We skip the details, but focus on the following theorems.

**Theorem A.14** (6.2.25 in Casella & Berger (2015)). *Let $\mathbf{x}_1, \cdots, \mathbf{x}_n$ be i.i.d. observations from an exponential family whose p.d.f. is given by $\Pr(\mathbf{x}|\boldsymbol{\eta}) = h(\mathbf{x})\exp(\boldsymbol{\eta}^\top \mathbf{T}(\mathbf{x}) - A(\boldsymbol{\eta}))$, then $\mathbf{T}(\mathbf{x}_1, \cdots, \mathbf{x}_n) = \sum_{i=1}^n \mathbf{T}(\mathbf{x}_i)$ is a complete statistic for $\boldsymbol{\eta}$ if support of $\boldsymbol{\eta}$ is open.*

**Theorem A.15** (Basu's Theorem, 6.2.24 in Casella & Berger (2015)). *If $\mathbf{T}(\mathbf{x})$ is a complete and minimal sufficient statistic, then $\mathbf{T}(\mathbf{x})$ is independent of every ancillary statistic.*

**Proof of Our Claim**  We now prove our claim made in the main paper. We copy Theorem 2 from the main paper here.

**Theorem 2.** *If $(\boldsymbol{\phi}_1, y_1), (\boldsymbol{\phi}_2, y_2), \cdots, (\boldsymbol{\phi}_n, y_n)$ are i.i.d. samples from the exponential family with $\mathbf{T}(\boldsymbol{\phi}, y) = \boldsymbol{\phi} \otimes \mathbf{e}_y$, then $\mathbf{T}((\boldsymbol{\phi}_1, y_1), \cdots, (\boldsymbol{\phi}_n, y_n)) = \sum_{i=1}^n \boldsymbol{\phi}_i \otimes \mathbf{e}_{y_i}$ is a minimal sufficient statistic independent of every ancillary statistic.*

*Proof.*    1. $\mathbf{T}((\boldsymbol{\phi}_1, y_1), (\boldsymbol{\phi}_2, y_2), \cdots, (\boldsymbol{\phi}_n, y_n))$ is sufficient by Thm. A.9.

2. $\mathbf{T}((\boldsymbol{\phi}_1, y_1), (\boldsymbol{\phi}_2, y_2), \cdots, (\boldsymbol{\phi}_n, y_n))$ is complete by Thm. A.14, since $\boldsymbol{\eta} \in \mathbb{R}^{m*n_{class}}$ is open.

3. Let

$$R = \frac{\Pr((\boldsymbol{\phi}_1, y_1), (\boldsymbol{\phi}_2, y_2), \cdots, (\boldsymbol{\phi}_n, y_n)|\boldsymbol{\eta})}{\Pr((\boldsymbol{\phi}_1', y_1'), (\boldsymbol{\phi}_2', y_2'), \cdots, (\boldsymbol{\phi}_n', y_n')|\boldsymbol{\eta})}$$

$$= \exp(\boldsymbol{\eta}^\top(\sum_{i=1}^n \boldsymbol{\phi}_i \otimes \mathbf{e}_{y_i} - \sum_{i=1}^n \boldsymbol{\phi}_i' \otimes \mathbf{e}_{y_i'}))$$

$\mathbf{T}((\boldsymbol{\phi}_1, y_1), (\boldsymbol{\phi}_2, y_2), \cdots, (\boldsymbol{\phi}_n, y_n))$ is also minimal by Thm. A.11, since $R$ is independent of $\boldsymbol{\eta}$ if and only if

$$\mathbf{T}((\boldsymbol{\phi}_1, y_1), \cdots, (\boldsymbol{\phi}_n, y_n)) = \mathbf{T}((\boldsymbol{\phi}_1', y_1'), \cdots (\boldsymbol{\phi}_n', y_n'))$$

4. $\mathbf{T}((\boldsymbol{\phi}_1, y_1), (\boldsymbol{\phi}_2, y_2), \cdots, (\boldsymbol{\phi}_n, y_n))$ is independent of every ancillary statistic by Thm. A.15.

$\square$

# B   Convergence Analysis

## B.1   Assumptions

We start by reviewing the notations and assumptions. Global communication rounds are denoted as $t \in \{0, \cdots, T\}$, local iterations as $e \in \{1, 2, \cdots, E\}$, and iterations of optimizing the global head as $k \in \{0, 1, 2, \cdots, K\}$. A local iteration step is denoted as $t(E + K) + e$, and a global gradient descent step is denoted as $t(E + K) + E + k$.

**Assumption 1** (Lipschitz Smooth). Each local objective function $\mathcal{L}(\mathcal{D}_c) := \mathcal{L}_c'$ is $L_1$-Lipschitz smooth w.r.t. the body $\boldsymbol{\theta}$, so the gradient of any local objective function is $L_1$-Lipschitz continuous. We will always omit $(\mathcal{D}_c)$ in $\mathcal{L}(\mathcal{D}_c)$ for simplicity.

$$||\nabla_{\boldsymbol{\theta}_c}\mathcal{L}_{t_1} - \nabla_{\boldsymbol{\theta}_c}\mathcal{L}_{t_2}||_2 \le L_1||\boldsymbol{\theta}_{t_1,c} - \boldsymbol{\theta}_{t_2,c}||_2^2, \forall t_1, t_2 > 0, \forall c \in S$$

, implying

$$\mathcal{L}_{t_1} - \mathcal{L}_{t_2} \le \langle \nabla_{\boldsymbol{\theta}_c}\mathcal{L}_{t_2}, \boldsymbol{\theta}_{t_1,c} - \boldsymbol{\theta}_{t_2,c}\rangle + \frac{L_1}{2}||\boldsymbol{\theta}_{t_1,c} - \boldsymbol{\theta}_{t_2,c}||_2^2$$

**Assumption 2** (Unbiased Gradient and Bounded Variance). Let $\xi_c \sim \mathcal{D}_c$ be a random batch of local datasets $\mathcal{D}_c$. The stochastic gradient $g_{t,c} = \nabla_{\boldsymbol{\theta}_c}\mathcal{L}(\xi_{t,c})$ is an unbiased estimator of the local gradient for each client.

$$\mathbb{E}_{\xi_c \sim \mathcal{D}_c}[g_{t,c}] = \nabla_{\boldsymbol{\theta}_c}\mathcal{L}_t$$

Its variance is bounded by $\sigma^2$.

$$\mathbb{E}[||g_{t,c} - \nabla_{\boldsymbol{\theta}_c}\mathcal{L}_t||_2^2] \le \sigma^2$$

**Assumption 3** (Lipschitz Smooth). Each local objective function $\mathcal{L}$ is $L_2$-Lipschitz smooth w.r.t. the head $\boldsymbol{\eta}$.

$$||\nabla_{\boldsymbol{\eta}}\mathcal{L}_{t_1} - \nabla_{\boldsymbol{\eta}}\mathcal{L}_{t_2}||_2 \le L_2||\boldsymbol{\eta}_{t_1} - \boldsymbol{\eta}_{t_2}||_2^2, \forall t_1, t_2 > 0$$

$$\mathcal{L}_{t_1} - \mathcal{L}_{t_2} \le \langle \nabla_{\boldsymbol{\eta}}\mathcal{L}_{t_2}, \boldsymbol{\eta}_{t_1} - \boldsymbol{\eta}_{t_2}\rangle + \frac{L_2}{2}||\boldsymbol{\eta}_{t_1} - \boldsymbol{\eta}_{t_2}||_2^2$$

**Assumption 4** (Heterogeneity). Let $\mathcal{D} = \cup_{c \in S}\mathcal{D}_c$ be the union of all datasets. For each client $c$, the angle between local gradients and global gradients w.r.t. the head $\boldsymbol{\eta}$ is bounded:

$$\exists H > 0, \langle \nabla_{\boldsymbol{\eta}}\mathcal{L}_t, \nabla_{\boldsymbol{\eta}}\mathcal{L}_t(\mathcal{D})\rangle \ge H||\nabla_{\boldsymbol{\eta}}\mathcal{L}_t||_2||\nabla_{\boldsymbol{\eta}}\mathcal{L}_t(\mathcal{D})||_2, \forall t > 0$$

We also assume the following norm is bounded:

$$\exists F > 0, 0||\nabla_{\boldsymbol{\eta}}\mathcal{L}_t||_2 \ge F||\nabla_{\boldsymbol{\eta}}\mathcal{L}_t(\mathcal{D})||_2, \forall t > 0$$

Assumption 4 essentially bounds the deviation between $\nabla_{\boldsymbol{\eta}}\mathcal{L}_t$ and $\nabla_{\boldsymbol{\eta}}\mathcal{L}_t(\mathcal{D})$. Since we showed local representations can be unified by fixing $\boldsymbol{\eta}$ during local training, we expect $\nabla_{\boldsymbol{\eta}}\mathcal{L}_t$ and $\nabla_{\boldsymbol{\eta}}\mathcal{L}_t(\mathcal{D})$ to be at worst perpendicular.

Additionally, we use the following notations: $\zeta$ is the step size of local training; $\beta$ is the step size of global head optimization.

## B.2 Key lemmas

From now on, the client subscription $c$ is omitted since the following lemmas are proven for any client.

**Lemma 1.** *Let Assumptions 1 and 2 hold. From the beginning of communication round $t$ to the last local update step, the loss function of any client is bounded as:*

$$\mathbb{E}[\mathcal{L}_{t(E+K)+E}] \leq \mathcal{L}_{t(E+K)} - (\zeta - \frac{L_1\zeta^2}{2}) \sum_{e=0}^{E-1} ||\nabla_{\boldsymbol{\theta}}\mathcal{L}_{t(E+K)+e}||_2^2 + \frac{L_1 E \zeta^2 \sigma^2}{2}$$

*Proof.*

$$\mathbb{E}_{\xi_{t(E+K)}}[\mathcal{L}_{t(E+K)+1}] \overset{(a)}{\leq} \mathbb{E}\left[\mathcal{L}_{t(E+K)} + \langle \nabla_{\boldsymbol{\theta}}\mathcal{L}_{t(E+K)}, (\boldsymbol{\theta}_{t(E+K)+1} - \boldsymbol{\theta}_{t(E+K)}) \rangle \right]$$
$$+ \mathbb{E}\left[\frac{L_1}{2}||\boldsymbol{\theta}_{t(E+K)+1} - \boldsymbol{\theta}_{t(E+K)}||_2^2\right]$$
$$= \mathcal{L}_{t(E+K)} + \mathbb{E}[\langle \nabla_{\boldsymbol{\theta}}\mathcal{L}_{t(E+K)}, -\zeta g_{t(E+K)} \rangle] + \frac{L_1\zeta^2}{2}\mathbb{E}[||g_{t(E+K)}||_2^2]$$
$$\overset{(b)}{=} \mathcal{L}_{t(E+K)} - \zeta||\nabla_{\boldsymbol{\theta}}\mathcal{L}_{t(E+K)}||_2^2 + \frac{L_1\eta^2}{2}\mathbb{E}[||g_{t(E+K)}||_2^2]$$
$$\overset{(c)}{\leq} \mathcal{L}_{t(E+K)} - \zeta||\nabla_{\boldsymbol{\theta}}\mathcal{L}_{t(E+K)}||_2^2 + \frac{L_1\eta^2}{2}(||\nabla_{\boldsymbol{\theta}}\mathcal{L}_{t(E+K)}||_2^2 + Var(g_{t(E+K)}))$$
$$\overset{(d)}{\leq} \mathcal{L}_{t(E+K)} - (\zeta - \frac{L_1\eta^2}{2})||\nabla_{\boldsymbol{\theta}}\mathcal{L}_{t(E+K)}||_2^2 + \frac{L_1\eta^2\sigma^2}{2}$$

(a): Assumption 1; (b): Assumption 2; (c) $\mathbb{E}[x^2] = (\mathbb{E}[x])^2 + Var(x)$; (d): Assumption 2.

By telescoping of $E$ steps, we have:

$$\mathbb{E}[\mathcal{L}_{t(E+K)+E}] \leq \mathcal{L}_{t(E+K)} - (\zeta - \frac{L_1\zeta^2}{2}) \sum_{e=0}^{E-1} ||\nabla_{\boldsymbol{\theta}}\mathcal{L}_{t(E+K)+e}||_2^2 + \frac{L_1 E \zeta^2 \sigma^2}{2}$$

$\square$

**Lemma 2.** *Let Assumptions 3 and 4 hold. After the global head learning at the server, the loss function of any client is bounded as:*

$$\mathcal{L}_{(t+1)(E+K)} \leq \mathcal{L}_{t(E+K)+E} - (\beta K H F - \frac{L_2\beta^2}{2}) \sum_{k=0}^{K-1} ||\nabla_{\boldsymbol{\eta}}\mathcal{L}_{t(E+K)+E+k}(\mathcal{D})||_2^2$$

*Proof.*

$$\mathcal{L}_{t(E+K)+E+1} \overset{(a)}{\leq} \mathcal{L}_{t(E+K)+E} + \langle \nabla_{\boldsymbol{\eta}}\mathcal{L}_{t(E+K)+E}, (\boldsymbol{\eta}_{t(E+K)+E+1} - \boldsymbol{\eta}_{t(E+K)+E}) \rangle$$
$$+ \frac{L_1}{2}||\boldsymbol{\eta}_{t(E+K)+E+1} - \boldsymbol{\eta}_{t(E+K)+E}||_2^2$$
$$\overset{(b)}{=} \mathcal{L}_{t(E+K)+E} + \langle \nabla_{\boldsymbol{\eta}}\mathcal{L}_{t(E+K)+E}, -\beta\nabla_{\boldsymbol{\eta}}\mathcal{L}_{t(E+K)+E}(\mathcal{D}) \rangle$$
$$+ \frac{L_1}{2}|| - \beta\nabla_{\boldsymbol{\eta}}\mathcal{L}_{t(E+K)+E}(\mathcal{D})||_2^2$$
$$\overset{(c)}{\leq} \mathcal{L}_{t(E+K)+E} - \beta H ||\nabla_{\boldsymbol{\eta}}\mathcal{L}_{t(E+K)+E}||_2 ||\nabla_{\boldsymbol{\eta}}\mathcal{L}_{t(E+K)+E}(\mathcal{D})||_2$$
$$+ \frac{L_1\beta^2}{2}||\nabla_{\boldsymbol{\eta}}\mathcal{L}_{t(E+K)+E}(\mathcal{D})||_2^2$$
$$\overset{(d)}{\leq} \mathcal{L}_{t(E+K)+E} - \beta||\nabla_{\boldsymbol{\eta}}\mathcal{L}_{t(E+K)+E}(\mathcal{D})||_2^2 HF + \frac{L_1\beta^2}{2})||\nabla_{\boldsymbol{\eta}}\mathcal{L}_{t(E+K)+E}(\mathcal{D})||_2^2$$

(a): Assumption 3; (b): global loss being evaluated on the aggregated data summary of all clients; (c): Assumption 4; (d): Assumption 4.

Again, by telescoping $K$ steps, we have

$$\mathcal{L}_{(t+1)(E+K)} \leq \mathcal{L}_{t(E+K)+E} - (\beta HF - \frac{L_2\beta^2}{2}) \sum_{k=0}^{K-1} ||\nabla_{\boldsymbol{\eta}} \mathcal{L}_{t(E+K)+E+k}(\mathcal{D})||_2^2$$

$\square$

### B.3 Theorems

We now prove Thm. 3-5 in the main paper.

**Theorem 3.** *Let Assumptions 1-4 hold. For any client, after each communication round, we have,*

$$\mathbb{E}[\mathcal{L}_{(t+1)(E+K)}] \leq \mathcal{L}_{t(E+K)} - (\zeta - \frac{L_1\zeta^2}{2}) \sum_{e=0}^{E-1} ||\nabla_{\boldsymbol{\theta}} \mathcal{L}_{t(E+K)+e}||_2^2 + \frac{L_1 E \zeta^2 \sigma^2}{2}$$

$$- (\beta HF - \frac{L_2\beta^2}{2}) \sum_{k=0}^{K-1} ||\nabla_{\boldsymbol{\eta}} \mathcal{L}_{t(E+K)+E+k}(\mathcal{D})||_2^2$$

*Proof.* Simply add Lemma 1 and Lemma 2. Take expectations of the random variable $\xi$ on both sides, and $\mathbb{E}[\mathcal{L}_{t(E+K)+E}]$ cancels out. $\square$

**Theorem 4.** *The expectation of the loss function $\mathbb{E}[\mathcal{L}]$ of any client monotonically decreases in every communication round when*

$$\zeta_{e'} < \frac{2(\sum_{e=0}^{e'} ||\nabla_{\boldsymbol{\eta}} \mathcal{L}_{t(E+K)+e}||_2^2)}{L_1(\sum_{e=0}^{e'} ||\nabla_{\boldsymbol{\eta}} \mathcal{L}_{t(E+K)+e}||_2^2 + E\sigma^2)}, e' \in \{0, 1, \cdots, E-1\}, \beta < \frac{2HF}{L_2}$$

*Proof.* Start from Thm. 3. We want

$$-(\zeta - \frac{L_1\zeta^2}{2}) \sum_{e=0}^{E-1} ||\nabla_{\boldsymbol{\theta}} \mathcal{L}_{t(E+K)+e}||_2^2 + \frac{L_1 E \zeta^2 \sigma^2}{2} < 0$$

$$-(\beta HF - \frac{L_2\beta^2}{2}) \sum_{k=0}^{K-1} ||\nabla_{\boldsymbol{\eta}} \mathcal{L}_{t(E+K)+E+k}(\mathcal{D})||_2^2 < 0$$

The inequalities are satisfied when

$$\zeta < \frac{2(\sum_{e=0}^{E-1} ||\nabla_{\boldsymbol{\eta}} \mathcal{L}_{t(E+K)+e}||_2^2)}{L_1(\sum_{e=0}^{E-1} ||\nabla_{\boldsymbol{\eta}} \mathcal{L}_{t(E+K)+e}||_2^2 + E\sigma^2)}, \beta < \frac{2HF}{L_2}$$

In practice, $\eta$ can be adjusted accordingly

$$\zeta_{e'} < \frac{2(\sum_{e=0}^{e'} ||\nabla_{\boldsymbol{\eta}} \mathcal{L}_{t(E+K)+e}||_2^2)}{L_1(\sum_{e=0}^{e'} ||\nabla_{\boldsymbol{\eta}} \mathcal{L}_{t(E+K)+e}||_2^2 + E\sigma^2)}, e' \in \{0, 1, \cdots, E-1\}$$

$\square$

**Theorem 5.** *Denote $\Delta = \mathcal{L}_0 - \mathcal{L}^*$, where $\mathcal{L}^*$ is the minimum value of $\mathcal{L}$. If $\zeta < min(\frac{2}{L_1}, \frac{2K\epsilon}{L_1(K+\sigma^2)}), \beta < \frac{2KHF}{L_2}$, denote $C_1 = \frac{2\beta HF - L_2\beta^2}{2\zeta - L_1\zeta^2} > 0$, then for any client, given any $\epsilon > 0$, after*

$$T > \frac{2\Delta}{E\zeta(\epsilon K(2 - L_1\zeta) - L_1\zeta\sigma^2)}$$

*communication rounds*

$$\frac{1}{TEK}\sum_{t=0}^{T-1}\sum_{e=0}^{E-1}\mathbb{E}[||\nabla_{\boldsymbol{\theta}}\mathcal{L}_{t(E+K)+e}||_2^2] + \frac{C_1}{TEK}\sum_{t=0}^{T-1}\sum_{k=0}^{K-1}\mathbb{E}[||\nabla_{\boldsymbol{\eta}}\mathcal{L}_{t(E+K)+E+k}(\mathcal{D})||_2^2] < \epsilon$$

*Proof.* We start from Thm. 3. Telescope considering the communication round from $t = 0$ to $t = T - 1$. Take expectation of $\xi$ on both sides, then,

$$\sum_{t=0}^{T-1}\mathbb{E}[\mathcal{L}_{(t+1)(E+K)}] \le \sum_{t=0}^{T-1}\mathbb{E}[\mathcal{L}_{t(E+K)}] - (\zeta - \frac{L_1\zeta^2}{2})\sum_{t=0}^{T-1}\sum_{e=0}^{E-1}\mathbb{E}[||\nabla_{\boldsymbol{\theta}}\mathcal{L}_{t(E+K)+e}||_2^2] + \frac{L_1 TE\zeta^2\sigma^2}{2}$$

$$- (\beta HF - \frac{L_2\beta^2}{2})\sum_{t=0}^{T-1}\sum_{k=0}^{K-1}||\nabla_{\boldsymbol{\eta}}\mathcal{L}_{t(E+K)+E+k}(\mathcal{D})||_2^2$$

Rearrange the inequality:

$$\frac{1}{TEK}\sum_{t=0}^{T-1}\sum_{e=0}^{E-1}\mathbb{E}[||\nabla_{\boldsymbol{\theta}}\mathcal{L}_{t(E+K)+e}||_2^2] + \frac{C_1}{TEK}\sum_{t=0}^{T-1}\sum_{k=0}^{K-1}\mathbb{E}[||\nabla_{\boldsymbol{\eta}}\mathcal{L}_{t(E+K)+E+k}(\mathcal{D})||_2^2]$$

$$\le \frac{\frac{2}{TE}\sum_{t=0}^{T-1}(\mathbb{E}[\mathcal{L}_{t(E+K)}] - \mathbb{E}[\mathcal{L}_{(t+1)(E+K)}]) + L_1\zeta^2\sigma^2}{K(2\zeta - L_1\zeta^2)}$$

$$\overset{(a)}{\le} \frac{2\Delta}{TEK(2\zeta - L_1\zeta^2)} + \frac{L_1\zeta\sigma^2}{K(2 - L_1\zeta)}$$

(a): $\sum_{t=0}^{T-1}(\mathbb{E}[\mathcal{L}_{t(E+K)}] - \mathbb{E}[\mathcal{L}_{(t+1)(E+K)}]) = \mathcal{L}_0 - \mathbb{E}[\mathcal{L}_{T(E+K)}] \le \Delta$. We want

$$\frac{2\Delta}{TEK(2\zeta - L_1\zeta^2)} + \frac{L_1\zeta\sigma^2}{K(2 - L_1\zeta)} < \epsilon$$

The inequality is satisfied when

$$T > \frac{2\Delta}{E\zeta(\epsilon K(2 - L_1\zeta) - L_1\zeta\sigma^2)}$$

$$\zeta < min(\frac{2}{L_1}, \frac{2K\epsilon}{L_1(K+\sigma^2)}), \beta < \frac{2HF}{L_2}$$

$\square$

## C  Differential Privacy

It has been shown that FL algorithms that share large model parameters do not prevent privacy attacks through weight manipulation, GAN-based reconstruction, and large model memorization effects (Boenisch et al., 2021; Mothukuri et al., 2021). We argue that FedLog, which shares summations of sufficient statistics only, avoid these pitfalls closely related to model parameter sharing. Since "addition" is a non-invertible function, malicious attackers cannot recover features of individual data points. Since the local architecture and weights of the feature extractor is never shared in anyway, malicious attackers should not be able to reconstruct the original inputs, even if they are given the features of individual data points. We acknowledge that this argument is merely intuitive, and sharing data summaries could pose other risks of privacy leakage. To further guarantee users' privacy formally, we now extend FedLog to be differentially private.

$(\epsilon, \delta)$-DP protects clients' privacy by adding noise to the shared information so that the adversaries cannot effectively tell if any record is included in the dataset (controlled by $\epsilon > 0$) at most times (controlled by $0 \leq \delta < 1$) (Kerkouche et al., 2021).

**Definition A.9.** A mechanism $M_{DP}$ satisfies $(\epsilon, \delta)$-DP if for any two datasets $\mathcal{D}, \mathcal{D}'$ that differ by only one record (i.e. $|(\mathcal{D} - \mathcal{D}') \cup (\mathcal{D}' - \mathcal{D})| = 1$), and for any possible output $O \in Range(M_{DP})$,

$$\Pr_{O \sim M_{DP}(\mathcal{D})} \left[ \log(\frac{\Pr[M_{DP}(\mathcal{D}) = O]}{\Pr[M_{DP}(\mathcal{D}') = O]}) > \epsilon \right] < \delta$$

Intuitively, $(\epsilon, \delta)$-DP guarantees that the inner log ratio, considered as the information loss leaked to the adversaries, is bounded by the privacy budget $\epsilon$ with probability $\delta$. Usually, $\epsilon \leq 1$ is viewed as a strong protection, while $\epsilon \geq 10$ does not protect much. The magnitude of noise needed is usually determined by $\epsilon, \delta$ and the sensitivity of the function $f$, of which the results ($f(\mathcal{D})$ the revealed information) need protection.

**Definition A.10.** The $L_p$ sensitivity of any function $f : \mathcal{D} \to \mathbb{R}^n$ is $L_p(f) = \max_{\mathcal{D}, \mathcal{D}'} ||f(\mathcal{D}) - f(\mathcal{D}')||_p$. $\mathcal{D}$ and $\mathcal{D}'$ differ by only one record.

A commonly used mechanism is to add Gaussian noise to $f(\mathcal{D})$:

**Theorem A.11** (Kairouz et al. (2015)). *For real-valued queries with sensitivity $L_2(f) > 0$, the mechanism that adds Gaussian noise with standard deviation $\sqrt{8k \ln(e + \epsilon/\delta)} L_2(f)/\epsilon$ satisfies $(\epsilon, \delta)$-differential privacy under $k$-fold adaptive composition, $\forall \epsilon > 0, \delta \in (0, 1]$.*

In FedLog, the only private information shared by clients is the summation of statistics $\mathbf{T}(\mathbf{\Phi}_c, \mathbf{y}_c)$, a vector of size $n_{class} * m$. Unfortunately, $L_2(\mathbf{T})$ is unbounded for standard deep neural networks $\tilde{\boldsymbol{\theta}}_c$, since the output features are usually unbounded. We need to clip the absolute values of the features to $b$, by simply adding an activation function to the last layer of the feature extractor

$$g(x) := \begin{cases} b, & \text{if } x > b \\ -b, & \text{if } x < -b \\ x, & \text{otherwise} \end{cases} \tag{14}$$

We now prove Thm. 7 from the main paper.

**Theorem 7.** *If the absolute value of features is clipped to $b$ and there are in total $k$ global update rounds, FedLog messages satisfy $(\epsilon, \delta)$-DP with additive Gaussian noise $\mathbf{T}'(\mathbf{\Phi}_c, \mathbf{y}_c) := \mathbf{T}(\mathbf{\Phi}_c, \mathbf{y}_c) + \mathcal{N}(\mathbf{0}, \sigma^2 \mathbf{I})$, where $\sigma = \sqrt{8k(1 + (m-1) * b^2) \ln(e + \epsilon/\delta)}/\epsilon$.*

*Proof.* We calculate the $L_2$ sensitivity as follows

$$L_2(\mathbf{T}) = \max_{\mathcal{D}, \mathcal{D}'} ||\mathbf{T}(\mathcal{D}) - \mathbf{T}(\mathcal{D}')||_2 \tag{15}$$

$$= \max_{\phi, y} ||\mathbf{T}(\phi, y)||_2 \tag{16}$$

$$= ||[1, b, b, \cdots, b, 0, 0, \cdots, 0]||_2 \tag{17}$$

$$= \sqrt{1 + (m-1) * b^2} \tag{18}$$

Eq. 15 equals to Eq. 16 due to our definition of neighbouring datasets (adding or removing one record). Eq. 16 equals to Eq. 17 since one of our features is always 1, and there are at most $m - 1$ other non-zero entries with maximum value of $b$.

Finally, we apply Thm. A.11 to get $\sigma = \sqrt{8k(1 + (m-1) * b^2) \ln(e + \epsilon/\delta)}/\epsilon$. □

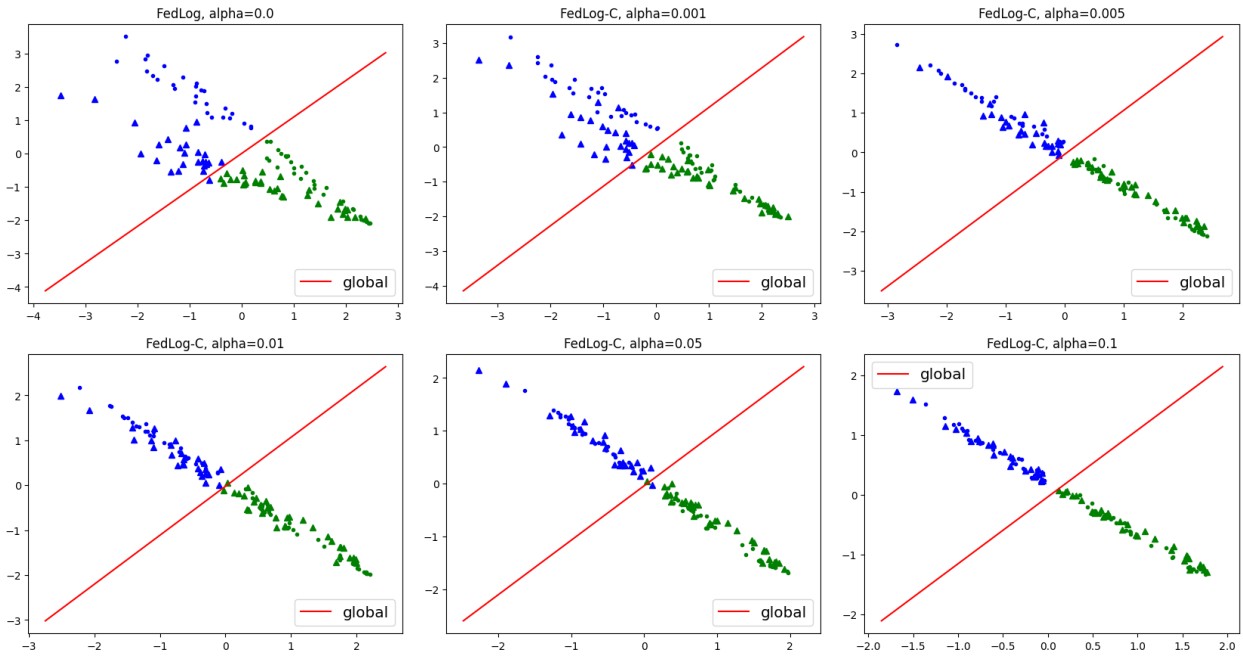

Figure A.1: Blue and green dots are local representations of class 0 and class 1. Circles and triangles show data distributed to client 0 or client 1. The red line is the global head.

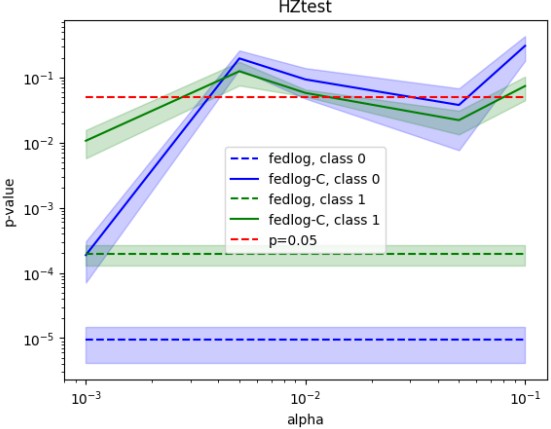

Figure A.2: HZtest results of the ablation study. Higher is better. Colored area show mean $\pm$ standard error.

Table A.1: $\Uparrow$ denotes significantly higher results with $p < 0.01$; $\Downarrow$ denotes significantly lower results with $p < 0.01$, and $\downarrow$ denotes $p < 0.05$.

| | MNIST 3 class/client | | MNIST 5 class/client | |
|---|---|---|---|---|
| | accuracy | comm cost | accuracy | comm cost |
| LG-FedAvg 1 | $95.82\pm0.12^{\Downarrow}$ | $11.91\pm4.66\text{Mb}^{\Uparrow}$ | $92.71\pm0.20^{\Uparrow}$ | $6.77\pm0.86\text{Mb}^{\Uparrow}$ |
| FedProto | $\mathbf{96.19\pm0.19}$ | $\mathbf{4.08\pm0.34Mb}$ | $\mathbf{93.01\pm0.26}^{\Uparrow}$ | $4.81\pm0.26\text{Mb}^{\Uparrow}$ |
| FedLog | $95.97\pm0.15^{\downarrow}$ | $5.96\pm0.88\text{Mb}^{\Uparrow}$ | $92.43\pm0.18$ | $6.12\pm0.63\text{Mb}^{\Uparrow}$ |
| FedLog-C | $\mathbf{96.13\pm0.10}$ | $\mathbf{4.08\pm0.42Mb}$ | $92.53\pm0.21$ | $\mathbf{3.18\pm0.39Mb}$ |

| | CIFAR10 3 class/client | | CIFAR10 5 class/client | |
|---|---|---|---|---|
| | accuracy | comm cost | accuracy | comm cost |
| LG-FedAvg 1 | $76.96\pm0.35^{\Downarrow}$ | $0.52\pm0.03\text{Gb}^{\Uparrow}$ | $66.30\pm0.36^{\Downarrow}$ | $0.44\pm0.02\text{Gb}^{\Uparrow}$ |
| FedProto | $78.37\pm0.31^{\Downarrow}$ | $0.38\pm0.03\text{Gb}^{\Uparrow}$ | $67.70\pm0.35^{\Downarrow}$ | $0.33\pm0.01\text{Gb}^{\Uparrow}$ |
| FedLog | $78.48\pm0.33^{\Downarrow}$ | $0.31\pm0.03\text{Gb}^{\Uparrow}$ | $68.06\pm0.39^{\Downarrow}$ | $0.28\pm0.02\text{Gb}^{\Uparrow}$ |
| FedLog-C | $\mathbf{79.18\pm0.31}$ | $\mathbf{0.22\pm0.03Gb}$ | $\mathbf{69.06\pm0.33}$ | $\mathbf{0.17\pm0.01Gb}$ |

| | CIFAR100 5 class/client | |
|---|---|---|
| | accuracy | comm cost |
| LG-FedAvg 1 | $72.43\pm0.29^{\downarrow}$ | $4.27\pm0.20\text{Gb}^{\Uparrow}$ |
| FedProto | $55.07\pm0.14^{\downarrow}$ | $6.03\pm0.14\text{Gb}^{\Uparrow}$ |
| FedLog | $\mathbf{74.29\pm0.25}$ | $2.75\pm0.05\text{Gb}^{\Uparrow}$ |
| FedLog-C | $\mathbf{74.48\pm0.28}$ | $\mathbf{2.52\pm0.05Gb}$ |

Table A.2: Results of the HZ-test on MNIST. Mean $\pm$ standard error is reported. $HZ'$ is the standardized test statistic.

| | LG-FedAvg 1 | FedProto | FedLog | FedLog-C |
|---|---|---|---|---|
| $HZ'$ | $-6080351 \pm 5785666$ | $-6311491 \pm 5672263$ | $-2221 \pm 380$ | $0.574 \pm 0.385$ |
| p-val | $0.0 \pm 0.0$ | $0.0 \pm 0.0$ | $0.0 \pm 0.0$ | $0.735 \pm 0.108$ |

## D Extra Experiments

### D.1 Ablation of the Auxiliary Loss in FedLog-C

We conducted the following experiment to study the effect of the auxiliary loss introduced in FedLog-C. We maintain a similar setup as the synthetic experiment, but increase the total number of data points to 120 for better visualization. We vary the weight of the auxiliary loss $\alpha$ from 0.001 to 0.1. As shown by Fig. A.1, local representations of both classes clearly form denser clusters as $\alpha$ increases. We further quantify how "Gaussian-like" the representations are with the Henze-Zirkler's Multivariate Normality Test (HZ-test) (Henze & Zirkler, 1990). Given a set of samples, HZ-test produces the p-value for the null hypothesis that such samples are from the same multivariate normal distribution. Thus, the larger the p-value is, the more likely the samples are from a Gaussian. If the p-value is larger than 5%, there is not enough evidence to reject the null hypothesis. Average results over 5 seeds are shown in Fig. A.2. It shows that with a proper $\alpha$, this auxiliary loss successfully forces the local representations to form Gaussian-like clusters, and this satisfies our assumption.

### D.2 Varied levels of heterogeneity

We report extra settings corresponding to different levels of heterogeneity in Table A.1. Results show FedLog-C still outperforms prior arts in most cases.

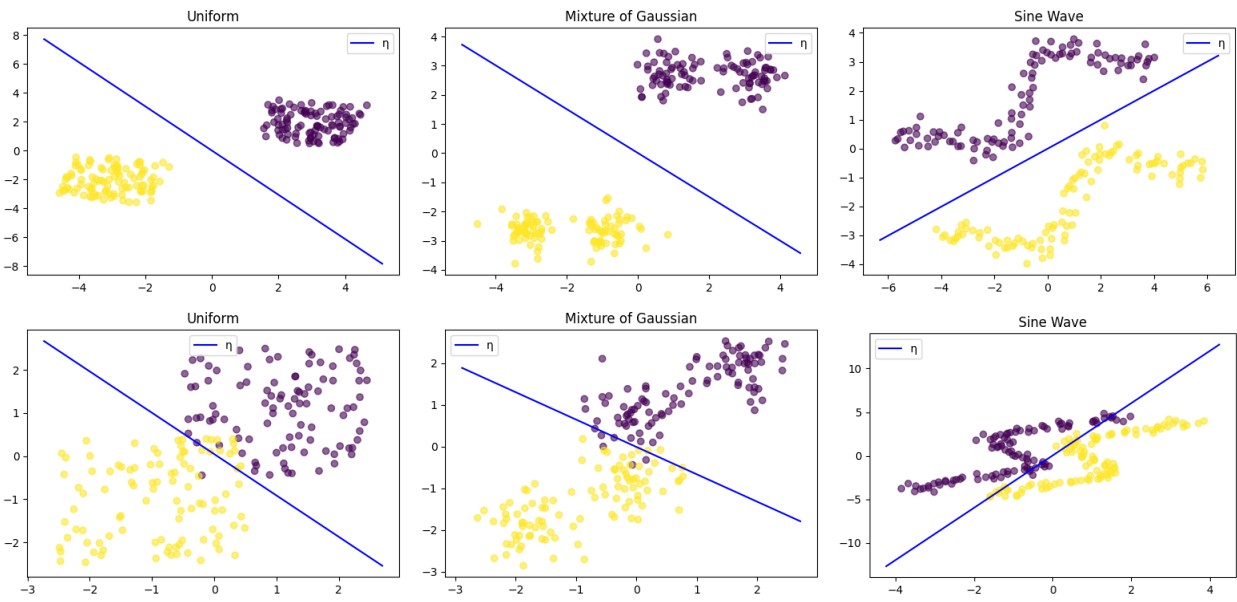

Figure A.3: Yellow and purple dots are data of two different classes. The blue line is the learned linear separator $\boldsymbol{\eta}$.

### D.3 P-values of MNIST

Due to floating-point precision limits, underflow often occurs when calculating the p-values of realistic datasets. We report the standardized test statistic $HZ' \in \mathbb{R}$ as an alternative. In HZ-test, $p = 1 - (1 - erf(HZ'))/2$, where $erf$ is the error function. Note the p-value increases monotonically as the statistic increases, and $p = 0.05$ when $HZ' \approx -1.163$. We also noticed that with high-dimensional representations (e.g. $\boldsymbol{\phi} \in \mathbb{R}^{100}$), HZ-test always fails even if $\boldsymbol{\Phi}$ is generated from a multivariate Gaussian distribution, probably also due to precision limits.

In Table A.2, we report $HZ'$ and p-values of all clients' representations of class 0, considering FedLog, FedLog-C, LG-FedAvg 1, and FedProto on MNIST. All the experimental details are maintained the same as in Sec. 4.2. Here $\boldsymbol{\phi} \in \mathbb{R}^{50}$. The result is the mean $\pm$ standard error of 10 random seeds. From the table, we can see that FedLog-C successfully shaped local representations to a Gaussian. FedLog itself also encourages more Gaussian-like clusters compared to the baselines.

### D.4 When the Gaussian assumption is not met

We design the following synthetic experiment to study how $\boldsymbol{\eta}$ reacts to non-Gaussian $\boldsymbol{\phi}$. We first sample 100 data points of two classes each. Data are sampled from either Uniform, Mixture of Gaussian, or a sine wave distribution. Then we instantly train $\boldsymbol{\eta}$ without learning a different representation. Results are shown in Fig. A.3. Yellow and purple dots are the samples of two classes, and the blue line is the linear separator induced by $\boldsymbol{\eta}$. We can see that FedLog can learn a proper $\boldsymbol{\eta}$, even if the Gaussian assumption is not met.

## E Experiment Details

Communication cost, flexible architecture, and differential privacy experiments are run on 1 NVIDIA T4 GPU with 16GB RAM. Celeba experiments are run on 1 NVIDIA A40 GPU with 48GB RAM. Training data are normalized and randomly cropped and flipped. The architectures of CNNs used are listed in Table A.3. Some

important hyperparameters are listed in Table A.4, A.5, and A.6. Most hyperparameters follow the experiment setting reported in LG-FedAvg. We make our code public on `https://github.com/rossyu/fedlog`, where further details can be found.

## F    Licences

Yann LeCun and Corinna Cortes hold the copyright of MNIST dataset, which is a derivative work from original NIST datasets. MNIST dataset is made available under the terms of the Creative Commons Attribution-Share Alike 3.0 license.

The CIFAR-10 and CIFAR-100 are labeled subsets of the 80 million tiny images dataset. They were collected by Alex Krizhevsky, Vinod Nair, and Geoffrey Hinton, made public at `https://www.cs.toronto.edu/~kriz/cifar.html`, the CIFAR homepage.

The CelebA dataset is available for non-commercial research purposes only. See `https://mmlab.ie.cuhk.edu.hk/projects/CelebA.html`, the CelebA homepage for the full agreement.

| MNIST | CIFAR10 | CIFAR100 |
|---|---|---|
| nn.Conv2d(1, 10, kernel_size=5) | nn.Conv2d(3, 6, kernel_size=5) | nn.Conv2d(3, 6, kernel_size=5) |
| F.max_pool2d(kernel_size=2) | nn.MaxPool2d(2,2) | nn.MaxPool2d(2,2) |
| nn.Conv2d(10, 20, kernel_size=5) | nn.Conv2d(6, 16, kernel_size=5) | nn.Conv2d(6, 16, kernel_size=5) |
| F.max_pool2d(kernel_size=2) | nn.MaxPool2d(2,2) | nn.MaxPool2d(2,2) |
| nn.Linear(320, 50) | nn.Linear(400, 120) | nn.Linear(400, 120) |
| nn.Linear(50, 10) | nn.Linear(120, 100) | nn.Linear(120, 100) |
| | nn.Linear(100, 10) | nn.Linear(100, 100) |

Table A.3: CNN architectures used in the communication cost experiment. Dropout layers and ReLu activation functions are omitted.

Table A.4: Hyperparameters used in communication cost experiments for MNIST.

| Algorithm | Hyperparameter | Value |
|---|---|---|
| FedLog | optimizer | Adam |
| | body learning rate | 0.001 |
| | head learning rate | 0.01 |
| | batch size | 10 |
| | local epochs | 5 |
| FedAvg | optimizer | Adam |
| | learning rate | 0.001 |
| | batch size | 10 |
| | local epochs | 5 |
| LG-FedAvg 1 | # global layers | 1 |
| | optimizer | Adam |
| | learning rate | 0.001 |
| | batch size | 10 |
| | local epochs | 5 |
| LG-FedAvg 2 | # global layers | 2 |
| | optimizer | Adam |
| | learning rate | 0.001 |
| | batch size | 10 |
| | local epochs | 5 |
| FedPer | optimizer | Adam |
| | learning rate | 0.001 |
| | batch size | 10 |
| | local epochs | 5 |
| FedRep | optimizer | Adam |
| | learning rate | 0.001 |
| | batch size | 10 |
| | body epochs | 5 |
| | head epochs | 10 |
| CS-FL | optimizer | Adam |
| | phase 1 learning rate | 0.001 |
| | phase 2 learning rate | 0.001 |
| | sparcity | 0.005 |
| | dimension reduction | 0.1 |
| | batch size | 10 |
| | local epochs | 5 |

Table A.5: Hyperparameters used in communication cost experiments for CIFAR10.

| Algorithm | Hyperparameter | Value |
|---|---|---|
| FedLog | optimizer | Adam |
| | body learning rate | 0.0005 |
| | head learning rate | 0.01 |
| | batch size | 50 |
| | local epochs | 1 |
| FedAvg | optimizer | Adam |
| | learning rate | 0.0005 |
| | batch size | 50 |
| | local epochs | 1 |
| LG-FedAvg 1 | # global layers | 1 |
| | optimizer | Adam |
| | learning rate | 0.0005 |
| | batch size | 50 |
| | local epochs | 1 |
| LG-FedAvg 2 | # global layers | 2 |
| | optimizer | Adam |
| | learning rate | 0.0005 |
| | batch size | 50 |
| | local epochs | 1 |
| FedPer | optimizer | Adam |
| | learning rate | 0.0005 |
| | batch size | 50 |
| | local epochs | 1 |
| FedRep | optimizer | Adam |
| | learning rate | 0.0005 |
| | batch size | 50 |
| | body epochs | 1 |
| | head epochs | 10 |
| CS-FL | optimizer | Adam |
| | phase 1 learning rate | 0.001 |
| | phase 2 learning rate | 0.01 |
| | sparcity | 0.0005 |
| | dimension reduction | 0.2 |
| | batch size | 10 |
| | local epochs | 1 |

Table A.6: Hyperparameters used in communication cost experiments for CIFAR100.

| Algorithm | Hyperparameter | Value |
|---|---|---|
| FedLog | optimizer | Adam |
| | body learning rate | 0.0005 |
| | head learning rate | 0.01 |
| | batch size | 50 |
| | local epochs | 3 |
| FedAvg | optimizer | Adam |
| | learning rate | 0.0005 |
| | batch size | 50 |
| | local epochs | 3 |
| LG-FedAvg 1 | # global layers | 1 |
| | optimizer | Adam |
| | learning rate | 0.0005 |
| | batch size | 50 |
| | local epochs | 3 |
| LG-FedAvg 2 | # global layers | 2 |
| | optimizer | Adam |
| | learning rate | 0.0005 |
| | batch size | 50 |
| | local epochs | 3 |
| FedPer | optimizer | Adam |
| | learning rate | 0.0005 |
| | batch size | 50 |
| | local epochs | 3 |
| FedRep | optimizer | Adam |
| | learning rate | 0.0005 |
| | batch size | 50 |
| | body epochs | 3 |
| | head epochs | 3 |
| CS-FL | optimizer | Adam |
| | phase 1 learning rate | 0.001 |
| | phase 2 learning rate | 0.01 |
| | sparcity | 0.0005 |
| | dimension reduction | 0.1 |
| | batch size | 10 |
| | local epochs | 1 |

