# OpenReview forum: "FedLog: Personalized Federated Classification with Less Communication and More Flexibility"
_TMLR — Accepted by TMLR_

### Review · Reviewer_hY5y · 2025-12-12

**Summary Of Contributions:**

The paper proposes a communication-efficient federated learning algorithm called FedLog. The method splits model parameters such that the server maintains a global head and clients train local feature extractors on this fixed head. After each round, the clients upload sufficient statistics (class-wise feature sums) which the server uses to update the head using Bayesian logistic regression.

The paper shows that by modeling the joint probability of the features and labels as an exponential family, the parameters of this distribution can be treated as the parameters of a linear classification head appended to the feature extractor. Furthermore, the paper shows that using Bayesian inference to update the head can be done with an update rule that only relies on summed feature statistics rather than individual examples.

**Audience:**

Yes

**Audience Explanation:**

It is important to explore methods beyond communicating model parameters for communication-efficient federated learning. As the paper shows, methods which communicate features can be highly efficient.

**Claims And Evidence:**

Yes

**Claims Explanation:**

The paper provides both theory and experiments to justify its claims. The connection between the exponential family and enabling the posterior update to be expressed in terms of the sufficient statistics is clear. The paper validates its claims on synthetic and image datasets and consistently shows improvement over other baselines.

**Requested Changes:**

A difference between FedLog and the baselines I am curious about is that FedLog freezes the head while the other baselines do not freeze parameters during local finetuning. Would it be worth running experiments to adapt the baselines by considering alternating updates e.g. see FedAlt in https://proceedings.mlr.press/v162/pillutla22a/pillutla22a.pdf.

It is unclear to what extent the features satisfy the mixture of Gaussians assumption, or if if such an assumption is necessary for good performance. The current experiment on synthetic data in Appendix D.1 is helpful, but it leaves a few more questions:
1. what are the p-values on other more realistic datasets?
2. what about other methods? does FedLog itself encourage more Gaussian-like clusters compared to baselines?
3. why does FedLog perform well, even though the p-values indicate that the data does not satisfy the assumption?

---

> ### Author Response · Authors · 2026-02-17
>
> > A difference between FedLog and the baselines I am curious about is that FedLog freezes the head while the other baselines do not freeze parameters during local finetuning. Would it be worth running experiments to adapt the baselines by considering alternating updates e.g. see FedAlt.
>
> - We have already compared to an instance of FedAlt, namely FedRep where the global model (body in this case) is fixed during local training. We report extra results with LG-FedAvg 1 + FedAlt (where the global head is frozen) in Sec 4.2. A slice of the table is pasted below.
>
> |                | MNIST                         | MNIST                      | CIFAR10                       | CIFAR10                       | CIFAR100                       | CIFAR100                           |   |
> |----------------|-------------------------------|----------------------------|-------------------------------|-------------------------------|--------------------------------|------------------------------------|---|
> |                | accuracy                      | comm cost                  | accuracy                      | comm cost                     | accuracy                       | comm cost                          |   |
> | LG-FedAlt      | 98.07$\pm$0.05$^\Downarrow$   | 3.91$\pm$0.27Mb$^\Uparrow$ | 86.54$\pm$0.3$^\Downarrow$    | 0.25$\pm$0.02Gb$^\Uparrow$    | 54.74$\pm$0.17$^\Downarrow$    | 4.47$\pm$0.16Gb$^\Uparrow$         |   |
> | **FedLog**   |  98.15$\pm$0.05$^\Downarrow$  | **3.18$\pm$0.31Mb**      |  87.08$\pm$0.22$^\Downarrow$  |   0.14$\pm$0.01Gb$^\Uparrow$  |   56.46$\pm$0.27$^\Downarrow$  | **2.38$\pm$0.09Gb$^\Downarrow$** |   |
> | **FedLog-C** | **98.41$\pm$0.07**          | **3.18$\pm$0.15Mb**      | **87.57$\pm$0.25**         | **0.11$\pm$0.01Gb**         | **56.78$\pm$0.26**           | 2.74$\pm$0.12Gb                    |   |
>
> > It is unclear to what extent the features satisfy the mixture of Gaussians assumption, or if such an assumption is necessary for good performance. The current experiment on synthetic data in Appendix D.1 is helpful, but it leaves a few more questions: 1. what are the p-values on other more realistic datasets? 2. what about other methods? does FedLog itself encourage more Gaussian-like clusters compared to baselines?
>
> - Due to floating-point precision limits, underflow often occurs when calculating the p-values of realistic datasets. We report the standardized test statistic $HZ' \in \mathbb{R}$ as an alternative. In HZ-test, $p = 1-(1-erf(HZ'))/2$, where $erf$ is the error function. Note the p-value increases monotonically as the statistic increases, and $p = 0.05$ when $HZ'\approx-1.163$. We also noticed that with high-dimensional representations (e.g. $\boldsymbol{\phi} \in \mathbb{R}^{100}$), HZ-test always fails even if $\boldsymbol{\Phi}$ is generated from a multivariate Gaussian distribution, probably also due to precision limits. We now report $HZ'$ and p-values of all clients' representations of class 0, considering FedLog, FedLog-C, LG-FedAvg 1, and FedProto on MNIST. All the experimental details are maintained the same as in Sec. 4.2. Here $\boldsymbol{\phi} \in \mathbb{R}^{50}$. The result is the mean $\pm$ standard error of 10 random seeds.
>
> |  | LG-FedAvg 1 | FedProto | FedLog | FedLog-C |
> |---|---|---|---|---|
> | HZ' | $-6080351\pm5785666$ | $-6311491\pm5672263$ | $-2221\pm380$ | $0.574\pm0.385$ |
> | p-val | $0.0\pm 0.0$ | $0.0\pm 0.0$ | $0.0\pm 0.0$ | $0.735\pm0.108$ |
>
> - From the table, we can see that indeed FedLog-C successfully shaped local representations to a Gaussian. FedLog itself also encourages more Gaussian-like clusters compared to the baselines.
>
> > 3. why does FedLog perform well, even though the p-values indicate that the data does not satisfy the assumption?
>
> - In addition to FedLog already encouraging Gaussian-like clusters, it seems FedLog is robust to different data distributions. We designed an additional synthetic experiment specifically to test how $\boldsymbol{\eta}$ would react to other distributions. Please see Appendix D.4 for the details.

---

### Review · Reviewer_5Ku1 · 2026-01-13

**Summary Of Contributions:**

This paper considers a federated version of deep representation learning and analyses the efficacy of a regime whereby only sufficient statistics are passed between clients and the central server. Specifically, each client has a body (a deep neural network) which learns a personalised compact feature representation, and a shared head is built to make class predictions. The method is developed under the assumption that the outputs of the body and the true classes are jointly in the exponential family, and passes sufficient statistics computed from these outputs to the server, rather than the raw data. Under this assumption, conjugate Bayesian updating for the head parameters is feasible and only succinct summaries of the data need be transmitted.

The authors propose a second method FedLog-C which augments the loss function used in training the local bodies, to encourage features which are in the exponential family (namely a mixture of Gaussians). This, as the authors note, bears similarity to CCVR (Luo et al. 2021) and FedPFT (Beitollahi et al. 2024) where Gaussian mixture models are fit to learned representations. The authors provide a convergence guarantee for FedLog and Differential Privacy guarantees.

Finally, the FedLog and FedLog-C methods are compared to baselines on synthetic and real datasets, with strong performance throughout.

**Audience:**

Yes

**Audience Explanation:**

It is within scope, engages with a timely and active research area and is well written.

**Broader Impact Concerns:**

N/A, the work is fairly foundational and seems not to have the potential to create any specific negative impacts.

**Claims And Evidence:**

Yes

**Claims Explanation:**

A method for federated representation learning is proposed, explained, and evaluated empirically. There are theoretical underpinnings in the form of convergence analysis and differential privacy guarantees, which appear sound.

**Requested Changes:**

-	A more detailed discussion of the differences between CCVR and FedPFT – perhaps in Section 3.4 once the proposed method is fully articulated. These methods seem, unless I have misunderstood, quite closely related?
-	What is the level of the significance tests mentioned in Section 4?
-	Sections 4.2-4.4 state the results of the experiments in detail, and the performance of the method is impressive, but it would be nice to see more justification as to why it works so well, like in the synthetic data experiments of Section 4.1.
-	It would be useful for practitioners to add some reflection, perhaps in the conclusion, on when FedLog versus FedLog-C ought to be used. It seems to be challenging to anticipate when the learned features will abide closely to a Gaussian distribution and not, and the justification in Section 3.4, while reasonable, is somewhat informal.

---

> ### Author Response · Authors · 2026-02-06
> **Clarification needed regarding the second question**
>
> Thank you for your feedback. We would like to request some clarification regarding the following question:
> - What is the level of the significance tests mentioned in Section 4?
>
> In Section 4, we have already reported the p-values for all statistical tests. In the table caption, we have indicated that ⇑ denotes significantly higher results with a p-value < 0.01, and ⇓ denotes significantly lower results with a p-value < 0.01. Is this the level of significance you are referring to, or are you requesting additional information or a different statistic?

---

> > ### Comment · Reviewer_5Ku1 · 2026-02-06
> > **Clarification**
> >
> > Apologies, I retract that question on re-reading. I'd missed the comment on the threshold for the p-values and got the idea that it was just significance or not being discussed without such a benchmark. This aspect is already clear, thanks.

---

> ### Author Response · Authors · 2026-02-17
>
> > A more detailed discussion of the differences between CCVR and FedPFT – perhaps in Section 3.4 once the proposed method is fully articulated. These methods seem, unless I have misunderstood, quite closely related?
>
> - We have edited Sec 3.4 to incorporate the following clarification. Although FedLog seems similar to CCVR and FedPFT, which also fit GMM to local features, FedLog differs fundamentally from this line of work in both the problem setting and the methodology: i) CCVR and FedPFT assume that a pre-trained body is available, and every client uses the same body for feature extraction, while FedLog and other baselines aim to train the whole model from scratch. For CCVR and FedPFT, local representations are not learned, but given for free; ii) CCVR and FedPFT simply sample additional data from the GMM, augment local datasets with these samples, and train the head in a non-Bayesian fashion, while FedLog constructs the head as canonical parameters and finds the posterior.
>
> > Sections 4.2-4.4 state the results of the experiments in detail, and the performance of the method is impressive, but it would be nice to see more justification as to why it works so well, like in the synthetic data experiments of Section 4.1.
>
> - We have added a few justifications to Sec 4.2-4.4, but ultimately Sec 4.1 is the justification for why it works so well: we use the synthetic data experiment to illustrate the key advantages of our design, which can be hard to isolate with real datasets.
>
> > It would be useful for practitioners to add some reflection, perhaps in the conclusion, on when FedLog versus FedLog-C ought to be used. It seems to be challenging to anticipate when the learned features will abide closely to a Gaussian distribution and not, and the justification in Section 3.4, while reasonable, is somewhat informal.
>
> - We have added the following to the conclusion. Based on experimental results, FedLog-C consistently outperforms FedLog in most settings. However, the results can be sensitive to the extra hyperparameter $\alpha$ introduced in FedLog-C. Thus, if hyperparameter searching is not feasible or differential privacy is needed, FedLog can be used instead. Otherwise, FedLog-C provides better accuracy and less communication cost.
>
> - We also have modified the justification in Sec 3.4 and conducted extra experiments in Appendix D.

---

### Review · Reviewer_Dx9w · 2026-02-05

**Summary Of Contributions:**

This paper introduces FedLog, a method for federated learning on classification problems that uses Bayesian inference and properties of exponential families to achieve low communication costs. The method is accompanied by theoretical guarantees (proof that the shared information is a minimal sufficient statistic, optimization analysis, differential privacy guarantee) and experimental evaluation against many baselines for synthetic data, image classification, and text classification.

**Additional Comments:**

Here are some questions that I had during the reading of this paper. I don't necessarily need the authors to respond with answers, but I think the paper would be improved if the answers to these questions are incorporated into the text.
- Is the model in 3.2 a novel design? Is it standard to parameterize an exponential family as in Eq (2) and (3)? Is it well known that this parameterization leads to an explicit expression of A? If so, can you add some references after Eq (2) and (3)?
- Why should the auxiliarly in Eq (13) enforce Gaussianity? It seems to just decrease variance? The empirical evidence that this happens (particularly in Fig A.2) is sufficient to accept that this loss is useful, but it would be helpful to say a sentence or two about how this loss is motivated or justified.
- Is it possible to achieve optimization guarantees and DP guarantees at the same time?  To clarify, I don't want you to add some new analysis that actually accomplishes this, I'm just curious if it could be done, and it might be useful to say a sentence or two about it.

**Audience:**

Yes

**Audience Explanation:**

Beyond the practical performance of the proposed method, the use of these Bayesian methods for representation learning in federated learning seems new. I think the FL community would be interested in learning about these techniques, and this work might inspire further utilization of these techniques in FL.

**Broader Impact Concerns:**

This work does not require any particular ethical discussion. The methods introduced by this paper are meant to achieve communication efficiency in federated learning, which has no more ethical implications than machine learning in general.

**Claims And Evidence:**

Yes

**Claims Explanation:**

The experimental performance of FedLog is strong: the proposed method often achieves both the highest test accuracy and the lowest (or near lowest) communication cost. As for the theory, each theoretical facet claimed as a contribution (proof of minimality, optimization guarantee, differential privacy guarantee) is accompanied by a theorem, and all proofs are (mostly) correct.

**Requested Changes:**

Major:
1. The optimization analysis in Section 3.5 has some issues that need to be addressed, discussed below:

    a. There are missing conditions on learning rate. The desired conclusion of Theorem 5 is not possible unless $\zeta \leq 2/(L_1 \sigma^2)$. Notice that the second to last equation on page 21 cannot be satisfied when $L_1 \zeta \sigma^2 >> \epsilon$.  Further, to get from this equation to the next one, you multiply both sides by $2 \zeta - L_1 \zeta^2$, which is negative unless $\zeta < 2/L_1$. So the theorem conclusion needs the additional condition $\zeta < \min( C_1/L_1, C_2 \epsilon/(L_1 \sigma^2) )$ for some constants $C_1, C_2$. Also, since you have to add these conditions for the conclusion to be valid, I recommend that you use them to simplify some expressions, for example $2 \zeta - L_1 \zeta^2$ can be replaced with the smaller $\zeta$ if $\zeta < 1/L_1$.

    b. The statement of Theorem 4 is not correct as stated. The proof guarantees that the expectation of the loss at each iteration is decreasing, not that the loss itself (which is a random variable) is decreasing. After this change, step (a) in the proof of Theorem 5 should be changed: the loss does not decrease monotonically, it only decreases in expectation, but this is sufficient for your purposes since the telescoping terms will cancel anyway.

    c. The heterogeneity assumption (Assumption 4) is not standard, as far as I can tell. This assumption eliminates the possibility of some $\eta$ that is a stationary point of a local objective, but not of the global objective, since in this case the ratio in Assumption 4 will not be bounded from below. In some ways, this is a strict assumption. Are there other works that make this assumption? Why is it necessary?

    d. The optimization analysis guarantees that the solution is an approximate stationary point only with respect to $\theta$, not $\eta$. I recommend changing the framing of the optimization guarantee so that G is included on the LHS, instead of appearing negatively on the RHS. This way you get a guarantee of stationarity in terms of both parameters $\theta$ and $\eta$.

    e. I don't think the optimization analysis really involves anything non-trivial. The analysis is essentially just a repetition of the standard analysis of SGD, though it requires a few extra steps since the authors consider this federated setup with the two separate variables $\theta$ and $\eta$. Still, I don't think that this result is much of a contribution.

    f. Can you clarify the notation $\mathcal{L}_c$ vs $\mathcal{L}_t$ in Section 3.5?  Do they mean different things or are you just using different variables as subscripts?

Minor:
1. When discussing that the communication is "minimal", I think it is important to point out that you mean "minimal" in thesense of "minimal sufficient statistics" as in Definition A.10. It would be helpful for the authors to disambiguate the use of the word "minimal" earlier in the manuscript, because in the context of federated learning this word is likely to refer to minimality in terms of some resource such as bits communicated or number of communication operations, rather than the statistical sense used here.
2. Some formatting suggestions:
- Use citep in places where citations appear directly in the sentence, for example in the first sentence and many places throughout the paper.
- Parentheses in Definition 6 or too small. Use \left( and \right).
- Fig A.2 should be smaller and centered.

---

> ### Author Response · Authors · 2026-02-17
>
> > The optimization analysis in Section 3.5 has some issues that need to be addressed.
>
> - We have updated the optimization analysis as requested. Please see Sec 3.5 and Appendix B. Assumption 4 is used in the proof of Lemma 2, so that we can have an upper bound of the deviation. It can be strict in general but feasible in FedLog, since we never update $\boldsymbol{\eta}$ locally and expect it not to fall into any local stationary point. $\mathcal{L}_t$ is short for $\mathcal{L}_t(\mathcal{D}_c)$ (as we stated we'd always omit $(\mathcal{D}_c)$ in Sec 3.5), which is the local loss $\mathcal{L}_c$ at a specific timestep $t$. Overall, we agree that there are no non-trivial steps introduced in the proof.  Note that analyzing the convergence rate of FedLog remains an important contribution.  In earlier feedback, we were asked to include such an analysis.  The fact that the analysis follows common steps in the analysis of the convergence of SGD should not be seen as a negative thing, but rather as positive since this makes the proof easier to follow.  The goal of this work is not to introduce new proof techniques for the sake of theory, but rather to introduce the FedLog algorithm.  Since FedLog is a new algorithm, there is a need to analyze its convergence and therefore leveraging existing proof techniques to derive the convergence rate is perfectly fine.  Finally, providing a theoretical analysis for a novel algorithm has often been listed as a contribution (e.g., FedProto).
>
> > When discussing that the communication is "minimal", I think it is important to point out that you mean "minimal" in the sense of "minimal sufficient statistics" as in Definition A.10. It would be helpful for the authors to disambiguate the use of the word "minimal" earlier in the manuscript, because in the context of federated learning this word is likely to refer to minimality in terms of some resource such as bits communicated or number of communication operations, rather than the statistical sense used here.
>
> - We have updated the paper to ensure every "minimal" is followed by "sufficient statistic" or is clearly defined throughout the paper.
>
> > Some formatting suggestions.
>
> - We have updated all the places where your suggestions apply.
>
> > Is the model in 3.2 a novel design? Is it standard to parameterize an exponential family as in Eq (2) and (3)? Is it well known that this parameterization leads to an explicit expression of A? If so, can you add some references after Eq (2) and (3)?
>
> - To the best of our knowledge, only Eq.(2) is similar to the choice of JEM (\textit{Grathwohl, W. et al. (2019). Your classifier is secretly an energy based model and you should treat it like one. }), an energy based model that also has a Kronecker product to retrieve the Softmax conditional likelihood. Eq.(3) is our novel design that leads to an explicit expression of A, while A in JEM is intractable. We have added a description in Sec 3.2.
>
> > Why should the auxiliarly in Eq (13) enforce Gaussianity? It seems to just decrease variance? The empirical evidence that this happens (particularly in Fig A.2) is sufficient to accept that this loss is useful, but it would be helpful to say a sentence or two about how this loss is motivated or justified.
>
> - The auxiliary in Eq (13) does not directly enforce Gaussianity. It does decrease variance and incentivize unimodal and light-tailed clusters, which are more likely to be Gaussian since the Gaussian distribution is unimodal and light-tailed. We have updated Sec 3.4 to incorporate this justification. As a side note, we tried using the p-value of the HZ-test as the auxiliary, but it was very unstable and performed worse than the current method.
>
> > Is it possible to achieve optimization guarantees and DP guarantees at the same time? To clarify, I don't want you to add some new analysis that actually accomplishes this, I'm just curious if it could be done, and it might be useful to say a sentence or two about it.
>
> - There are recent works that can be applied (\textit{Khah, S. et al. (2025). Differentially Private Clipped-SGD: High-Probability Convergence with Arbitrary Clipping Level.}). It seems that with DP, convergence is not guaranteed but only with a high probability (e.g. Thm 4.1).

---

> > ### Comment · Reviewer_Dx9w · 2026-02-18
> >
> > Thanks for the updates, I think that the revised manuscript is definitely an improvement. On my concerns about the optimization results, I believe that now the results are logically correct, but I still have some reservations about their significance and the validity of the assumptions. On the significance side, I stand by what I said before, that the convergence analysis is a minor variation on the standard convergence analysis of SGD, which is not really a meaningful contribution. However, I am sympathetic to the fact that it is standard to provide convergence analysis for proposed algorithms (as requested by a previous reviewer, according to the authors), so perhaps a minor variation on a standard proof is better than no proof at all. In a similar vein, the heterogeneity assumption (Assumption 4) appears very non-standard to me and I still don't understand why this condition is necessary. Since the proof is a variation of the SGD analysis, can we not use a similar assumption as the standard SGD analyses, such as bounded gradient dissimilarity as in e.g. [1]? I believe that removing the non-standard assumption would improve the quality of the optimization analysis, however I won't insist that the authors make the change, because in my mind the real contribution of the paper is in the algorithm design and experimental results.
> >
> > [1] Woodworth, Blake E., Kumar Kshitij Patel, and Nati Srebro. "Minibatch vs local sgd for heterogeneous distributed learning." Advances in Neural Information Processing Systems 33 (2020): 6281-6292.

---

### Author Response · Authors · 2026-02-17
**Revised Paper Uploaded**

We sincerely thank the reviewers for their valuable feedback. All requested changes have been incorporated into the revised paper, with important modifications documented and highlighted in teal in the supplementary file changes_colored.pdf.

---

### Decision · Action_Editor_BxVu · 2026-03-17

**Recommendation:** Accept as is

**Audience:**

Yes

**Audience Explanation:**

Using properties of exponential families to improve federated learning algorithms is both interesting and novel, this is a good fit for TMLR's audience.

**Claims And Evidence:**

Yes

**Claims Explanation:**

The proposed method is backed by analysis and experiments which were judged accurate by the reviewers.